

# Spatial distribution of gaseous pollutants (NO₂, SO₂, NH₃, HNO₃ and O₃) in Abidjan, Cote d'Ivoire

Julien Bahino[1], Véronique Yoboué[1], Corinne Galy-Lacaux[2], Marcellin Adon[1], Aristide Akpo[3], Sékou Keita[1], Cathy Liousse[2], Eric Gardrat[2], Christelle Chiron[2], Money Ossohou[1], Sylvain Gnamien[1] and Julien Djossou[3].

[1] Laboratoire de Physique de l'Atmosphère, Université Félix Houphouët-Boigny, Abidjan, BPV 34, Cote d'Ivoire
[2] Laboratoire d'Aérologie, UMR 5560, Université Toulouse III Paul Sabatier and CNRS, 31400, Toulouse, France
[3] Laboratoire de Physique du Rayonnement, Université d'Abomey Calavi, Benin

*Correspondence to*: Julien Bahino (julienbahino@gmail.com)

**Abstract.** This work is part of the DACCIWA FP7 program (Dynamics-Aerosol-Chemistry-Cloud Interactions in West Africa) in the framework of the work package 2 « Air Pollution and Health ». This study aims to characterize urban air pollution levels through the measurement of NO₂, SO₂, NH₃, HNO₃ and O₃ in Abidjan, the economic capital of Cote d'Ivoire. Gases measurements are performed using INDAAF (International Network to study Deposition and Atmospheric chemistry in AFrica) passive samplers exposed in duplicate for two weeks periods. We performed an intensive measurement campaign in Abidjan from December 15th, 2015 to February 16th, 2016 during the dry season. Twenty-one sites were selected in the district of Abidjan to be representative of various anthropogenic and natural sources of air pollution in the city. We collected 672 samples of gas during this period. Results from the intensive campaign show that gas concentrations are strongly linked to pollution sources nearby and show a high spatial variability on the different sites of Abidjan. However, three gases present relative higher levels of concentrations at all the sites: NH₃, NO₂ and O₃. NH₃ average concentrations vary between 9.1 ± 1.7 ppb at a suburban site and 102.1 ± 9.1ppb at a domestic fires site. NO₂ mean concentration vary from 2.7 ± 0.1 ppb at a suburban site to 25.0 ± 1.7 ppb at an industrial site. We measured the two highest O₃ concentration on the two coastal sites located in the southeast of the city with average concentration of 19.1 ± 1.7 ppb and 18.8 ± 3.0 ppb respectively for Gonzagueville and the Felix Houphouet-Boigny international airport. The SO₂ average concentration never exceeds 7.2 ± 1.2 ppb at all the sites with 71.5 % of the sampling sites presenting concentrations ranged between 0.4 ppb and 1.9 ppb. The HNO₃ average concentration is comprised between 0.2 ppb and 1.4 ppb. All these results were combined with meteorological parameters to provide the first mapping of gaseous pollutants at the scale of the district of Abidjan using the geostatistical analysis (ArcGIS software). Spatial distribution results emphasize the importance of the domestic fires source and the significant impact of the traffic emissions at the scale of the city. In addition, we propose in this work a first overview of gaseous SO₂ and NO₂ concentrations at the scale of several African cities from literature compared to our measurements. The daily SO₂ standard of WHO is exceeded in most of the cities reported in the overview where concentrations range from 0.2 µg.m⁻³ - 3662 µg.m⁻³. Annual NO₂ concentrations ranged from 2 µg.m⁻³ - 175 µg.m⁻³ are lower than the WHO threshold. As a conclusion, this study constitutes an original database to characterize urban air pollution and a first attempt toward a spatialization of the pollution levels at the scale of the metropolis of Abidjan. This work should draw the attention of the African public authorities to the necessity of air quality monitoring network in order to (1) to define national standards and to better control the pollutants emissions and (2) to investigate the impact on the health of the growing population of developing African countries.



## 1 Introduction

For several years, urban areas have experienced a deterioration in air quality and an increase in health and environmental impacts. Several scientific studies have shown that a large number of premature deaths, respiratory and cardiovascular diseases are related to air pollution (Brook et al., 2004; Murray, 2001; Pope III et al., 2002; Pope III and Dockery, 2006; WHO, 2006).

According to the World Health Organization estimation in 2012, 11.6 % of the deaths in the world were associated with outdoor and indoor air pollution. It represents nearly 6.5 million of deaths a year among which 3 million deaths were attributable only to ambient air pollution. 88 % of these deaths occurred in low- and middle-income countries (WHO, 2016). According to the OECD Development Centre, 700,000 premature deaths were linked to air pollution in Africa in 2013 (Roy Rana, 2016). Anthropogenic activities are the main sources of emissions of gaseous and particulate pollutants into the air and their

concentration measured in the urban atmosphere (Kampa and Castanas, 2008). This problem has led the scientific community to address the issue in order to attract the attention of political authorities. Several studies have been carried out in the major cities of Europe, Asia and northern America. These studies have shown the emergency of taking action to reduce emission of pollutant into the atmosphere in order to mitigate health and environmental impacts. Air quality networks have been set up in the major cities of United States of America and Europa to inform the public authorities and the population in real time. In

Africa where researchers have found that air pollution causes more premature deaths per year than either unsafe drinking water or malnutrition, there is almost no quality network (Roy Rana, 2016). However, very few studies on air pollution are carried out on the African continent and particularly on the West African region and in South Africa (Norman et al., 2007). The West African region has experienced an economic upturn these last years characterized by an economic growth estimated at 4.2 % in 2015 with a prospect of 6.3 % in 2016 (UNECA, 2016). A strong industrialization, an increase of trade, the improvement

of transports and the building of modern infrastructure drive this growth. Most of the economic activity of the African countries are concentrated in the cities. In consequence, we notice a strong demographic explosion of cities favoured by massive rural exodus and by important migrations of the West African populations (Denis and Moriconi-Ebrard, 2009). This intense economic activity associated with the rapid population growth, the strong urbanization and the perpetual uncontrolled expansion of the city involve increasingly important anthropogenic emissions of gaseous and particulate pollutants. This causes

a significant deterioration of air quality that could alter the health of populations and ecosystems (Fourn and Fayomi, 2006; Norman et al., 2007). The sanitary consequences induced by air pollution are now a cause for concern in the population. However, very few qualified dataset on levels of pollutants concentrations are available for African cities to sensitize populations and authorities in order to promote appropriate measures to reduce pollution. Through some research programs and projects such as AMMA (African Monsoon Multidisciplinaty Approach), INDAAF and POLCA (POLlution des Capitales

Africaines), some experimental studies have been carried out in some African capitals and rural areas such as in Bamako, Dakar, Yaounde, and Amersfoort (Adon et al., 2016; Josipovic et al., 2010; Liousse and Galy-Lacaux, 2010; Lourens et al., 2011; Val et al., 2013). The DACCIWA (Dynamics-Aerosols-Chemistry-Clouds Interactions in West Africa) European program follows the framework of the previous cited programs. The measurements implemented within the framework of DACCIWA are mainly focused on the South of the West African region. DACCIWA has conducted extensive fieldwork in

SWA (South West Africa) to collect high-quality observations, spanning the entire process chain from surface-based natural and anthropogenic emissions to impacts on health, ecosystems, and climate (Knippertz et al., 2015). WP2 work package of the DACCIWA program aims to link emission sources, air pollution and health impacts in terms of lung inflammation and related diseases over representative differentiated urban sources in South West Africa: traffic, domestic fires and waste burning in Abidjan (Cote d'Ivoire), and two-wheel vehicle traffic in Cotonou (Benin). The strategy in WP2 is based on a multidisciplinary

approach (physics, chemistry, toxicology, epidemiology, modelling…). Experimental results of WP2 rely both on short intensive campaigns and also on mid-term monitoring measurements on the different urban selected sites. Three urban supersites to represent the major urban sources previously cited have been selected for the mid-term monitoring from December 2014 to March 2017. This study presents results of the intensive campaign performed from December 15[th], 2015 to February





16th, 2016. This work, part of the WP2, aims to characterize urban air pollution in the metropolis of Abidjan through the measurement of gaseous pollutants ($NO_2$, $SO_2$, $NH_3$, $HNO_3$ and $O_3$) concentrations. Twenty-one sampling sites distributed throughout the district of Abidjan were selected to be instrumented with passive samplers of gases to obtain from the observations the spatial distribution of each gaseous pollutant. This study allows to compare the contribution of each source

of pollution and to compare the levels of concentration obtained in Abidjan with those of other cities in Africa and other developing countries.

## 2 Experimental design

Abidjan, the economic capital of Cote d'Ivoire is a cosmopolitan city of sub-Saharan Africa and one of the most important
city in West Africa. Abidjan is located on the Ebrie Lagoon on the Gulf of Guinea in the south-east coast of Cote d'Ivoire. According to the National Statistics Institute, the mean annual population growth in Abidjan is about 2.6 %. The population has increased from 3.1 million in 1998 to 4.7 million in 2014 (NSI, 2015). With a population of about five million, Abidjan is the second most-populated African city, surpassed only by the Nigerian city of Lagos. Thus, Abidjan could be considered as a metropolis in West Africa with a rapid growth all over the surrounding municipalities. Today, the autonomous district of
Abidjan is composed by ten municipalities in the city (Abobo, Adjamé, Attécoubé, Cocody, Koumassi, Marcory, Plateau, Port-Bouët, Treichville and Yopougon) and three neighboring municipalities (Anyama, Bingerville and Songon). The cumulated areas of these municipalities represent an area of 2119 km$^2$, i.e. 0,66 % of the country (Yao-Kouassi, 2010) against 580 km$^2$ in 1990 (Kopieu, 1996). Abidjan is a port city and a dynamic economic center, not only for the country, but also for all the West African sub-region. The port of Abidjan, at the first rank in West Africa, contains 70 % of the country's industries and
constitutes the main maritime facade for the hinterland countries (Burkina Faso, Niger and Mali). Abidjan shelters the main part of the companies of all the sectors of economic activity of the country. It represents 60 % of the Gross Domestic Product (GDP) of Cote d'Ivoire estimated according to the World Bank at 31.76 billion US dollars in (World Bank, 2016). This growth is mainly due to the port activity, to the creation of new companies and to the permanent installation of subsidiaries of international companies in the district of Abidjan. This intense economic activity required the creation of new industrial zones.
The main sectors that support this economic growth are real estate, road transport, road building, manufacturing and mining industries.

A port city, the 6th metropolis of the continent and economic powerhouse for West Africa, Abidjan has experienced in recent years strong demographic and economic growth which emphasize different environmental issues such the air pollution. This is one of the main issue in West Africa capitals driven by explosive unplanned growth of urban conglomerations and faced to
unregulated emissions.

To estimate the level of air pollution in the city, it is important to take into account most of the anthropogenic activities that may constitute sources of pollution. As a consequence, several measurement sites have been selected to spatially represent the different municipalities of the district of Abidjan.

### 2.1 Sampling sites

In this work, we developed an experimental strategy to build a network of measurement sites. The initiation of this measurement network was based on the conclusion of a preliminary report published in 2006 by the Canadian Tecsult firm ordered by the ministry of environment of Cote d'Ivoire. This study had envisaged the installation of an air quality monitoring network in the District of Abidjan. The report identified and fully described 50 potential measurement sites that finally have not been implemented. In this work, we take benefit of this first report to consider 15 sites. Our study uses the main criteria
that guided the choice of sites and the objectives of DACCIWA program. These criteria are as follow: (1) to consider the population density of each municipalities as a factor of potential high sources of gaseous pollutant, (2) to obtain a representative



spatial coverage of the 13 municipalities of the district of Abidjan and (3) to document the main sources of air pollution identified within the framework of the program DACCIWA (domestic fires, road traffic and waste burning) (Knippertz et al., 2015). Figure 1 presents the mapping of the 21 selected sites referenced from $A_1$ to $A_{21}$ and the major sources of pollution nearby in the district of Abidjan. Table 1 presents the details of each site by giving the name of the site, its geographical

coordinates and the land use denomination. 10 sampling sites are classified as traffic sites ($A_1$ to $A_8$, $A_{10}$, $A_{14}$) and 5 of them present a combination of traffic and a second major source of pollution ($A_4$, $A_7$, $A_8$, $A_{10}$, $A_{14}$). Among these sites, $A_1$ is the DACCIWA supersite representative of traffic.  3 sites are representative of industrial areas ($A_9$ to $A_{11}$) and 4 sites of residential and popular district ($A_7$, $A_8$, $A_{12}$ and $A_{13}$). 4 sites have been selected as suburban and one as a green area ($A_{17}$ to $A_{21}$). Finally, two DACCIWA supersites studied over a two years period (2015-2017) in terms of local emission sources are part of this

study to be representative of the domestic fires ($A_{16}$) and waste burning ($A_{15}$).

**2.2 Sampling and analysis**

**2.2.1 Sampling procedure**

Concentrations of gaseous pollutants ($NO_2$, $SO_2$, $HNO_3$, $NH_3$ and $O_3$) were determined in ambient air of the district of Abidjan using the INDAAF (http://indaaf.obs-mip.fr) passive samplers method (Adon et al., 2010). These passive samplers developed

by Laboratory of Aerology (LA) in Toulouse (France) using the work of (Ferm, 1991) were tested and qualified in the framework of the INDAAF project for the great African ecosystems over about fifteen years (Adon et al., 2010). Passive samplers have also been used for the measurement of gaseous pollutants concentration in urban areas during the POLCA (Pollution of African Capitals) program (Adon et al., 2016). Passive samplers can be used for indoors and outdoors monitoring in all environment (Salem et al., 2009) and provide good results (Gorecki and Namiesnik, 2002). They are suitable to

investigate the spatial distribution of gaseous pollutants and establish the air quality monitoring network (Carmichael et al., 2003; Cox, 2003; Cruz et al., 2004). INDAAF passive samplers are small, silent and reliable. They don't need electricity and are based on the property of molecular diffusion of gases and species specific collection on impregnated filter to each pollutant measured (Galy-Lacaux et al., 2001). The calculation of gaseous pollutants concentrations in the air depends on the physical characteristics of the passive samplers, the duration of exposure and the meteorological parameters. We calculate gaseous

pollutant average concentration in ppb using the following formula given by (Adon et al., 2010) Eq. (1):

$$C_{amb}(ppb) = \frac{(L/A).X.R.T}{t.D.P},$$    (1)

$C_{amb}$ (ppb) represents the concentration of the considered gaseous pollutant in ppb, X is the number of gas molecules trapped on the cellulose filter (µmol), R= 0.08206 L atm mol$^{-1}$ K$^{-1}$. T is the average temperature during the sampling period (K), P is the mean atmospheric pressure during the sampling period, D is the molecular diffusion coefficient of gas in the air, t is the

sampler's exposure duration (s) and the ratio L/A= 47.5 m$^{-1}$ is the air resistance coefficient for the INDAAF passive sampler (Adon et al., 2010).

The passive samplers have been prepared by the Laboratory of Aerology in Toulouse and dispatched in Abidjan, Cote d'Ivoire from December 15th, 2015 to February 16th, 2016 during the dry season. For each set of passive samplers set, blank samples are kept sealed to be used as field blanks. Some blanks are kept at the Laboratory and others sent to the sites. Passive samplers

were exposed bi-monthly on the 21 measurement sites of the district of Abidjan to represent integrated two weeks period over the two months studied period. Passive samplers were mounted on stainless steel rails in duplicate and placed at a height of at least 2.5 meters above the ground. So, on each rail, we installed eight passive samplers with different colours for the different gases: two white passive samplers to measure ammonia ($NH_3$), two black for nitric acid ($HNO_3$) and sulfur dioxide ($SO_2$), two grey for nitrogen dioxide ($NO_2$) and two grey/black for ozone ($O_3$) (Fig. 2). The reproducibility of the INDAAF passive

samplers was found to be 20 %, 9.8 %, 14.3 %, 16.6 % and 10 % for $HNO_3$, $NO_2$, $NH_3$, $SO_2$ and $O_3$ respectively. Detection





limits for each trace gas were calculated from field blanks filters and found to be $0.07 \pm 0.03$ ppb for $HNO_3$, $0.2 \pm 0.1$ ppb for $NO_2$, $0.7 \pm 0.2$ ppb for $NH_3$, $0.05 \pm 0.03$ ppb for $SO_2$ and $0.1 \pm 0.1$ ppb for $O_3$ (Adon et al., 2010). Passive samplers are then exposed for 2 weeks before being replaced by the new ones. Once removed, passive samplers are kept refrigerated before being shipped to the LA for analysis.

### 2.2.2 Chemical analysis

 Each passive sampler contains a cellulose filter that is specifically pre-impregnated with a selective chemical solution to react with the studied gases molecules. All the details concerning the impregnation of chemical solutions and the associated reactions on filters, as far of the detailed analytical IC procedure are given in Adon et al. (2010). To determine and calculate the
concentration of studied gaseous pollutants, ionic concentrations of ammonium ($NH_4^+$), nitrate ($NO_3^-$), sulfate ($SO_4^{2-}$) and nitrite ($NO_2^-$) are determined by ion chromatography (IC) after desorption of the filters in 10 mL of 18 MΩ deionized water by ultrasonic stirring (15 min). The volume of desorption is 5 mL for the ammonia passive sampler. The ion chromatography system used for this study has been extensively described by (Adon et al., 2010; Galy-Lacaux and Modi, 1998; Hodgkins et al., 2011). To check the reliability of the IC analytical results, the chemistry laboratory of LA of Toulouse participates twice a
year since 1996 to the quality control inter-comparison program organized by WMO/GAW (World Meteorological Organization/ Global Atmospheric Watch) (Laouali et al., 2012). Results of the WMO/GAW quality assurance program for the year 2016 shows that analytical precision is estimated to be $\pm 5$ % for all ions (http://www.qasac-americas.org/)**.** Over the study period, 672 passive samplers have been exposed and 2 were damaged. 670 samples and 48 blanks (12 for each gas) were analysed. 24 blanks have been kept in the laboratory and 24 other blanks were sent in the same condition with exposed passive
samplers. Average concentration for each trace gas were calculated from field blanks and found to be $0.37 \pm 0.02$ ppb for $NO_2$, $1.4 \pm 0.18$ ppb for $NH_3$, $0.09 \pm 0.00$ ppb for $HNO_3$, $0.01 \pm 0.00$ ppb for $SO_2$ and $0.08 \pm 0.04$ ppb for $O_3$. All samples are collected in duplicate. The average concentration of two samples is used in all the case except when contamination of one sample is suspected and when the average concentration is under the detection limit. 4 samples (0.6 %) were removed from the whole database. The final database represents 666 samples i.e 99.4 % of the analysed samples.

### 2.3 Meteorological data

Meteorological parameters used for this study were collected at the site ($A_6$). This site is located at the weather station of the Félix Houphouet-Boigny international airport. This weather station is managed by ASECNA (Agency for Safety of Air Navigation in Africa and Madagascar) and provides meteorological data (Temperature, relative humidity, wind velocity and direction, pressure…) for air navigation in the district of Abidjan. The collected data for this study are direction and wind
velocity, temperature and relative humidity. Daily variations of temperature and relative humidity during the studied period (December 15th, 2015 to February 16th, 2016) at the site $A_6$ are presented in Fig. 3. Daily average temperature ranged between 25 °C and 29 °C. Mean relative humidity was about 78 % with a maximum of 92 % during the study period. This period was particularly dry with only two rainy days and a total cumulative rainfall of 32 mm. The direction and velocity of wind during the study period on the site $A_6$ are represented on the wind rose of Fig. 4.The study of the hourly wind of Abidjan on the study
period shows the predominance of southwest (SW) and north- northeast (NNE) winds directions. Wind velocity presents values ranged between 1 and 9 meters per second with an average of $2.75 \pm 0.6$ m.s$^{-1}$. The observed meteorological parameters are characteristic of the major dry season in Cote d'Ivoire.

### 2.4 Spatial analysis

In this study, we interpolated the concentrations measured at the 21 sites to build spatial distribution of gases concentrations
of the entire city of Abidjan using geostatistical analysis (GIS). The mapping tool is ArcGIS software in its version 10.2.2.





The features were projected using World Geodetic System (WGS_1984) UTM_Zone_30N (Universal Transverse Mercator). Two interpolation methods are frequently used to study spatial distribution of pollutants. There are IDW (Inverse Distance Weighted) and kriging. The general formula for both interpolators is formed as a weighted sum of the data Eq. (2):

$$\hat{Z}(S_0) = \sum_{i=1}^{N} \lambda_i Z(S_i),\qquad\qquad(2)$$

$\hat{Z}(S_0)$ is the predicted concentration at $S_0$ location and is calculated as a linear weighted sum of N observations surrounding the predicted location. Z ($S_i$) is the measured concentration at the $S_i$ location, $\lambda_i$ is an unknown weight for measured concentration at the $S_i$ location, $S_0$ is the prediction location and N is the number of measured concentration. In IDW, the weight, $\lambda_i$ depends only on the distance between the measurement points and the prediction location. For kriging method, the weights $\lambda_i$ are based not only to the distance but also on the overall spatial arrangement of the sampling sites. IDW computes

predicted concentrations at the location as a weighted average of neighbouring measured concentrations (Ibrahim et al., 2012; Rivera-Gonzalez et al., 2015). This method provides a spatial distribution of pollutants much closer to the reality. IDW method showed better similarity between measured and interpolated concentrations of sulfur dioxide ($SO_2$) and nitrogen dioxide ($NO_2$) (Jha et al., 2011). Kriging is the method of interpolation deriving from regionalized variable theory. It depends on expressing spatial variation of the property in terms of the variogram, and it minimizes the prediction errors which are themselves

estimated (Oliver and Webster, 1990). According to kriging, we calculated for each studied pollutant, the coefficient of correlation between the interpolated concentrations and observed concentrations. Fig. 5a shows the regression between interpolated and observed concentration of $O_3$, $SO_2$, $HNO_3$ and $NO_2$ using ordinary kriging. For this study, results showed that the map produced extremely good predictions of monitored pollution level for average concentration on the 21 sampling sites during the dry season. Kriging gives best correlation coefficient for $O_3$ followed by $SO_2$, $HNO_3$ and $NO_2$ with respectively (r

= 0.97, 0.92, 0.92, 0.74). The critical value of the Pearson's correlation coefficient at the threshold of 5 % is r = 0.42. The correlation coefficient obtained for $NH_3$ is r = 0.02. Kriging tends to overestimate the ammonia concentrations. Almost 70 % of the predicted values are above the observed concentrations. The absolute difference between the interpolated values and the observed values is between 3 and 69 ppb for 19 sampling sites out of 21. Extremely high concentrations of $NH_3$ were obtained on the sampling sites $A_{14}$ and $A_{16}$ with 67.7 ppb and 102.1 ppb respectively. These concentrations correspond to measurements

carried out within 5 m of the source of pollution. By removing these extreme concentrations of $NH_3$ from the database during spatial interpolation, and using ordinary kriging, we recalculated the coefficient of correlation. Fig. 5b shows the regression between interpolated and observed concentration of $NH_3$ using ordinary kriging. The correlation coefficient obtained between observed and interpolated concentrations increased from r = 0.02 to 0.75. Regarding the high correlation coefficient between observed concentrations and interpolated concentrations, we chose ordinary kriging to represent the spatial distribution of $O_3$,

$NO_2$, $SO_2$, $HNO_3$ and $NH_3$.

### 3 Results and discussion

During this study from December 15th, 2015 to February 16th, 2016, we collected 672 samples of the studied gas ($NO_2$, $SO_2$, $HNO_3$, $NH_3$, $O_3$). We compile all the quality controlled passive samplers results to produce an original database for the district of Abidjan. Concentrations of gaseous pollutants at each sampling location for each two weeks were determined as the mean

of two duplicates. The mean concentrations in ppb presented in Table 2 for the 21 sites are calculated from the mean of the four two-weeks periods sampled. We first discuss together these mean gaseous concentrations with the results we obtained of spatial distribution at the scale of the district of Abidjan. Then we will discuss gaseous pollutant concentrations by anthropogenic sectors of activity. A comparison of the average concentrations of pollutants at the scale of the District of Abidjan with other cities of Africa and Asia is also given. Finally, we finished this work by an overview of $NO_2$ and $SO_2$ air

monitoring studies in Africa in order to assess how reported data compare to WHO air guidelines.





### 3.1 Spatial distribution of gaseous pollutants in Abidjan

In order to present the spatial distribution of gaseous pollutants under investigation, the center of the District of Abidjan is divided into rectangle of 35 km x 30 km between 5°22' N and 5°26' N, and 4°04' W and 4°07' W. Maps of each pollutant are presented using a colour scale from blue (low concentration) to red (high concentration in ppb) (Fig. 6 to Fig. 10). Black lines

present administrative boundaries. Sampling sites are represented by black points and referenced with an ID code from $A_1$ to $A_{21}$. Green lines on the spatial distribution map of $NO_2$ represent the main road network.

#### *Nitrogen dioxide (NO₂)*

The spatial distribution of $NO_2$ concentrations in the district of Abidjan is represented on Fig. 6. This map indicates that $NO_2$ average concentrations during this dry season vary between 2.7 ppb and 25 ppb. The highest values were obtained at the Tri

postal site ($A_{11}$) (25 ppb). This site is representative both of the port and industrial activities. The second highest concentration is measured at the traffic site $A_5$ with a mean concentration of 23.9 ppb. It represents the signature of the pollutants emissions of the administrative center of Abidjan characterized by traffic jams every working day. Only those 2 sites have an average concentration higher than the WHO (World Health Organization) guideline value (40 µg.m$^{-3}$ or 21 ppb annual mean). Sites $A_1$, $A_8$, and $A_9$, representative of traffic and industrial sites present comparable mean concentrations ranged between 19.9 ppb

and 21 ppb. $NO_2$ concentrations decrease rapidly when we leave the city center of Abidjan to the periphery where the road network is less important and the main roads drawn in green on Fig 6. Similar results were found by Gilbert et al. (2003) in Montréal and Rodes and Holland (1981) in Los Angeles. Lowest values were measured on the suburban sites $A_{19}$ (5.4 ppb) and $A_{18}$ (2.7 ppb). The average $NO_2$ concentrations measured in Abidjan are usually most important when the sampling sites are located near an important road network. Traffic appears to be one of the main source of $NO_2$ emission in Abidjan. In 2014,

more than 82 % of vehicles in circulation in Cote d'Ivoire tumbled into the district of Abidjan (Kouadio, 2014). The national agencies of urban transport (AGETU (AGEnce des Transports Urbains) and SONATT (SOciété NAtionale des Transports Terrestres)) counted near than 423,587 vehicles of any types in 2010 (Goore Bi, 2011). Most of these vehicles are second-hand and have more than 20 year old (Ministry of Transport, 2012). These vehicles that no longer respect the standards of current air quality in European Union (E.U) are most used because of their moderate price. The engines of these vehicles reject black

smokes resulting from an incomplete combustion. Emissions of these black smokes are more important for the rush hours during traffic jams caused by the narrowness and the state of degradation of roads. The emission factor of black carbon (BC), in g/kg fuel for the use of diesel and gasoline in road traffic are respectively 3.1 ± 1.9 and 0.62 ± 0.5. These emission factors for African developing countries were measured in Cote d'Ivoire and Benin (Keita et al, 2017, pers.com). Beside the road traffic, the industrial sector seems to be another great contributor to highest measured $NO_2$ concentration. Significant

concentration levels around 20.9 ppb are measured on the three industrials sites ($A_{11}$, $A_9$, and $A_{10}$). Abidjan has currently three industrial zones occupying a surface of 885 ha and the fourth is under construction. The biggest one is located at the municipality of Yopougon (645 ha) followed by the industrial zones of Vridi in the municipality of Port-Bouet (120 ha) and of Koumassi (120 ha) (ICI, 2014). The industrial zone of Vridi ($A_{11}$) where the highest level concentration of $NO_2$ were measured (25 ppb) includes several activities related to the harbour area, shipyards, oil terminals, chimneys, thermal power

station and processing and storage of petroleum products factories. Except oil and shipping industries, most of these activities are found in the other industrial zones. Other activities such as cement manufacturing, chemical industries and plastic production are also present. World Health Organization (2006) air quality guidelines and the work done by Krzyzanowski and Cohen, (2008) identify combustion processes in stationary sources (heating, power generation) and in mobile sources (internal combustion engines in vehicles and ships) as major sources of anthropogenic emission of nitrogen oxides into the atmosphere.




### *Sulfur dioxide (SO$_2$)*

Figure 7 shows the spatial distribution of the SO$_2$ concentrations in Abidjan. SO$_2$ concentrations varied from a minimum of 0.4 ppb in the suburban site (A$_{20}$) to the maximum of 7.2 ppb measured at the (A$_3$) traffic site (Table 2). The gap between these two values is mainly due to the location of the sites in the district of Abidjan and the related activities around. The highest

concentrations were generally obtained on the five traffic sites (A$_3$, A$_2$, A$_4$, A$_{19}$, A$_5$) and on the A$_1$ suburban site with respective value of 7.2 ppb, 6.8 ppb, 4.7 ppb, 3.4 ppb, 3.1 ppb, and 2.6 ppb. We notice a decrease of SO$_2$ concentrations from the Northwest of the city where the main roads (Northern Highway and East Road) and the industrial zone of Yopougon (A$_9$) are located to the Southeast of Abidjan near the Atlantic Ocean. The majority of the sampling sites (71.5 %) presents concentrations lower than 2 ppb. As for NO$_2$, the highest SO$_2$ concentrations are observed in areas of dense traffic and in industrial zones. It

should be noted that refining companies located in the industrial zone of Vridi (A$_{11}$) don't have yet desulfurization unit. This kind of production unit could constitute an important source of SO$_2$ (Rodriguez and Hrbek, 1999; Tam et al., 1990). The sulfur content of the fuel produced and marketed in Cote d'Ivoire is higher than 2000 part per million (ppm) (CCAC and UNEP, 2016). This content is more than 40 times higher than the standards applied in most of the developed countries (50 ppm or below). There is also an intensive charcoal production through biomass burning in the northwest of the city. The large rubber

plantations located in this area are regularly reconverted into charcoal using traditional ovens. According to the study NAMA (Nationally Appropriate Mitigation Action) on sustainable charcoal in Cote d'Ivoire, the charcoal production increased by 22 % between 400,850 t in 2003 to 488,128 t in 2012. During the same period, firewood production increased from 8,699,979 m$^3$ to 9,034,617 m$^3$ (UNDP, 2015). Charcoal (47 %) and firewood (35 %) are the main sources of energy for households in urban area in Cote d'Ivoire. These products resulting from the incomplete combustion of carbon compounds are also important

sources of SO$_2$ emission (Andreae, 1991).

### *Ammonia (NH$_3$)*

NH$_3$ is the most abundant alkaline gas in the atmosphere. The main source of NH$_3$ emissions is biomass burning including biofuel combustion, agriculture, animal husbandry, NH$_3$-based fertilizer, industrial processes and vehicular emission (Behera et al., 2013). In urban areas, domestic fires, traffic and industrial activity are generally considered as major sources of NH$_3$

(Perrino et al., 2002; Sapek, 2013; Sutton et al., 2000; Whitehead et al., 2007). In West African capitals, domestic fires and biomass burning are the main source of NH$_3$ emission (Liousse et al., 2014). Anthropogenic emissions of NH$_3$ for the domestic fires source in Bamako (Mali) and Dakar (Senegal) have been estimated for 2005 to be $1.6 \times 10^{-3}$ TgNH$_3$ yr$^{-1}$ and $7.0 \times 10^{-4}$ TgNH$_3$ yr$^{-1}$ respectively (Adon et al., 2016). In Abidjan NH$_3$ is also the gas that is found in a greater proportion on all the sampling sites (Table 2). Figure 8 shows the spatial distribution of NH$_3$ in Abidjan. The domestic fires site (A$_{16}$) has the highest

concentration of NH$_3$ (102.1 ± 9.1 ppb). This site is a long term instrumented monitoring site of the DACCIWA project located near a traditional smokehouse of food products. It was selected to study the concentrations of pollutants emitted by domestic fires to the source. The second most important concentration of NH$_3$ was measured at the site A$_{14}$ located on the east of the city and characterized by domestic fires and the residential sector (67.7 ± 8.3 ppb). It should be noted that these 2 sites presenting very high concentrations corresponds to measurements performed less than 5 m from the source of pollution and

were not integrated into the realization of the spatial distribution map of NH$_3$. (section 2.4). Generally, NH$_3$ concentrations are ranged from 9.1 ± 1.7 ppb to 44.0 ± 2.2 ppb. High concentrations about 36.6 ± 4.1 ppb are measured in the west and the northeast of the city where most of the domestic fires, residential and suburban sites are located. One part of the Songon municipality located in the south-west of Abidjan between sites A$_{18}$ and A$_{19}$ regroups a set of poultry and pig farms because of the availability of space and its rural character (Yapi-Diahou et al., 2011). Golly and Koffi-Didia (2015) showed the

existence of 63 modern poultry farms and 9 modern pig farms in this region. The liquid manure resulting from these breedings is one of the main sources of NH$_3$ and nitrogenous compounds emission (Degré et al., 2001). Total NH$_3$ emissions in France



in 2007 were estimated at 382 kt with 15 % of these emissions were due to poultry dejections (Gac et al., 2007). In the global NH$_3$ anthropogenic source budget, (Bouwman et al., 2002) indicates that animal excreta source represents 21.7 Mt yr$^{-1}$ of the total 53.6 Mt yr$^{-1}$. In addition to the various breedings, this area is full of large industrial and individual plantations of hevea and banana. These plantations often make excessive use of chemical fertilizers rich in urea. The degradation of urea leads to

the formation of NH$_3$ (Degré et al., 2001). At the east of Abidjan, we also observe on the waste burning site (A$_{15}$) high concentration of NH$_3$ (39.1 ± 2.1 ppb). This site is located in the village of Akouedo and classified as an uncontrolled dump site of garbage (Kouame et al., 2006). With 153 ha, it receives all the vegetable, animal and industrial waste of the city of Abidjan since 1965. This huge waste is estimated at about 550,000 t per year (Kouame et al., 2006). The decomposition of the various waste could constitute an important source of NH$_3$ emission as shown by our measurements. To summarize, NH$_3$

concentrations measured in the city of Abidjan represent a mixture of several sources (waste degradation, animals, agriculture…) associated with constant contribution of the domestic fires source.

### *Nitric acid (HNO$_3$)*

HNO$_3$ and NH$_3$ are important trace constituents of the atmosphere (Cadle et al., 1982). HNO$_3$ is believed to be a major end product in oxidation of gaseous nitrogen compounds. It is one of the mineral acid contributor to the acid rain (Cogbill and

Likens, 1974). Figure 9 shows the spatial distribution of HNO$_3$ in Abidjan during the dry season. The study of the spatial distribution of HNO$_3$ in Abidjan showed that the concentrations are ranged between 0.2 ± 0.0 ppb and 1.4 ± 0.3 ppb. Approximately half of the sampling sites have concentration between 1.1 ± 0.3 ppb and 1.4 ± 0.3 ppb. These sites are mainly located in the center and in the west of the city. The most important HNO$_3$ concentrations was measured at a suburban site A$_{17}$ (1.4 ± 0.3 ppb), a waste burning site A$_{15}$ and an industrial area A$_9$ (1.3 ± 0.2 ppb), a residential site. A$_{13}$, a domestic fires site

A$_{16}$ and a popular district A$_7$ (1.2 ± 0.3). HNO$_3$ appears to be emitted from several sources in the district of Abidjan. This could be explained by the fact that gaseous HNO$_3$ is the result from several chemical transformation processes of NH$_3$ and NOx (NO$_2$ + NO) guided by presence of oxidants such as H$_2$O$_2$, O$_3$ and OH radical (Seinfeld and Pandis, 2016; Kumar et al., 2004). Thus industrial activities, traffic, domestic fires, waste burning which may emit these different chemical compounds can contribute to levels of HNO$_3$ in atmosphere.

### *Ozone (O$_3$)*

O$_3$ is one of the most important contaminants in urban areas and has been associated with adverse effects on human health and natural environment (Hagenbjörk et al., 2017). O$_3$ is not emitted directly by primary sources, but formed in the lower atmosphere from photochemical interactions of gaseous precursors composed of nitrogen oxides (NO$_x$) and volatile organic compounds (VOCs) (Duan et al., 2008; WHO, 2006). At a specific location, the level of O$_3$ concentration depends on emission

to its precursors (VOCs and NOx), the long-range transport of O$_3$ and meteorological parameters (Hagenbjörk et al., 2017). Two chemical regimes are associated with the rate of ozone production. The NOx-saturated or (VOC-limited) regime and the NOx-limited regime. In urban areas with high NOx concentrations (NOx-saturated) and a low ratio VOCs/NOx, the ozone production rate is generally low and concentrations are in the order of few ppb. In some places, high concentrations can be observed, due to the recirculation of atmospheric air mass and specific meteorological conditions. When we leave the city

center, in the peri-urban areas, we notice an increase of COV/NOx ratio and a high O$_3$ production because of the absence of major sources of NOx, the dilution of air mass and the presence of biogenic VOCs. We measured O$_3$ concentration at different location in the district of Abidjan. Figure 10 shows the spatial distribution of ozone in the district of Abidjan during the dry season. Generally, we notice the presence of O$_3$ throughout the city. The levels of concentration increase considerably from the west to the east of Abidjan. The most important O$_3$ concentrations were observed in the two coastal sampling sites A$_6$

(Gonzagueville) and A$_{20}$ (Airport-FHB) located in the southeast of the city with 19.1 ppb and 18.8 ppb respectively. These





two sites on the eastern outskirts of the city are characterized by a monthly sunshine ranging between 117 hours and 224 hours (Messou et al., 2013) and by a high relative humidity in the air due to the presence of water vapor from the marine spray. The OH radical is the major oxidant of the atmosphere. It directly controls lifetime of VOCs and NOx (Sadanaga et al., 2012). It ensures the conversion of NO to $NO_2$ and is therefore responsible for the net production of ozone (Camredon and Aumont,

2007). We also measured significant concentrations at two other peri-urban sites in the east of the city:  the waste burning site of the village of Akouedo ($A_{15}$) and the suburban site of Bingerville ($A_{17}$) with $O_3$ concentrations of 17.55 ppb and 16.76 ppb respectively. On the traffic sites ($A_1$, $A_2$, $A_7$, $A_9$) located near the Banco forest, ozone concentration was found to be between 11.3 ppb and 12.9 ppb. The lowest $O_3$ concentration in Abidjan was measured in the north of the city on the site $A_4$ (5.09 ppb).

**3.2 Mean gaseous concentrations of the main anthropogenic activities in Abidjan and comparison with other cities of**
**developing countries**

The study of the spatial distribution of gaseous pollutants in Abidjan reveals different anthropogenic contributions to the measured levels of pollution. Therefore, we have tried to investigate the relation between the concentrations levels measured for each pollutant and the potential emission of each anthropogenic activity. Sampling sites were grouped by activity sectors source of pollution. Six different groups were formed. The main activities sources of pollution have been identified in section

3.1 to be traffic, industrial, residential, domestic fires, waste burning and suburban. For each sector, we calculated the mean concentration of each gaseous pollutant (Fig. 11). The analysis shows that the industrial sector mainly explains the highest measured $NO_2$ concentrations with an average concentration of 20.9 ± 2.8 ppb. However, domestic fires, traffic and residential sites present very comparable levels of $NO_2$ concentrations of 17.9 ± 0.3 ppb, 17.8 ± 4.7 ppb and 16.1 ± 2.4 ppb respectively compared to industrial sites. It should be noted that residential and domestic fires sites are strongly influenced by road traffic

in Abidjan. The identified residential and domestic fires sites are never more than 10 meters far from roadways. This could explain the near-similar concentrations levels observed between traffic, domestic fires and residential sites. Waste burning (12.0 ± 0.0 ppb) and suburban (7 ± 2.4 ppb) sites present lower concentration of $NO_2$. Nevertheless, these concentrations are not negligible and can be due, on one hand, to the limited traffic of the garbage collection vehicles and also on the other hand to the circulation of minibus called "gbaka" which connects the city centre to the suburbs.

Average $O_3$ concentrations are higher at the waste burning (17.6 ± 0.0 ppb) and suburban (12.2 ± 4.3 ppb) sites. All these sites are located far from the city centre. This result confirms the $O_3$ production far from the sources of primary pollutants and confirms the study of Klumpp et al. in 2006 on ozone pollution at urban and suburban areas. Traffic and industrial sites show comparable mean $O_3$ concentrations with 10.2 ± 1.3 ppb and 10.2 ± 1.5 respectively. $NH_3$ concentrations are significantly higher at domestic fires sites with an average concentration two to four times higher than for other sites (84.9 ± 17.2 ppb). This

very high value highlights the use of biomass burning (firewood and charcoal) as a source of energy by most of households of Abidjan. The high standard deviation is due to the location of 2 of the domestic fires sites near the source while the other domestic fires sites are much further away from the source. The others groups of sites representative of waste burning (39.1 ppb) and traffic sources (27.2 ± 3.1 ppb) present $NH_3$ concentrations lower than 40 ppb. Peri-urban agriculture and intensive livestock in the Abidjan suburbs are mainly responsible for the suburban areas $NH_3$ concentration (22.6 ± 10.3 ppb). This value

is close to concentrations measured at residential (21.9 ± 5.2 ppb) and industrial (20.7 ± 0.7 ppb) sites. Atmospheric $HNO_3$ in Abidjan is much more the fact of the waste burning sites with an average concentration of 1.3 ± 0.0 ppb. All other sources of pollution have $HNO_3$ concentration ranging from 0.6 ± 0.3 ppb to 0.9 ± 0.2 ppb. The road traffic sector is the main contributor to the $SO_2$ pollution levels measured in Abidjan (4.2 ± 2.2 ppb). This result seems to be a good indicator of the quality of fuels used in Abidjan. In countries without stringent fuel policies such as Cote d'Ivoire, diesel sulphur content is generally important.

In order to place the concentrations of gaseous pollutants in Abidjan in a sub-regional and international context, we have listed in Table 3 some experimental studies using passive sampler to have a comparable measurement technique and also a




comparable temporal integration. Table 3 presents gaseous pollutants concentrations in African cities and in some industrialized cities of emerging Asian countries. The average concentration of $NO_2$ observed during the studied period at the different sites in Abidjan is generally lower than the WHO's annual average threshold (21 ppb or 40 $\mu g.m^{-3}$). However, the average concentration observed at the industrial sites is about 20.9 ppb and varies between 17.7 ppb and 25 ppb. These values

move closer to the threshold and sometimes exceed it at some of the sampling sites. These concentrations can affect health of most of vulnerable populations, such as children and elder (Barnett et al., 2005; Pandey et al., 2005). In Abidjan, we can assume that the risk exist because of the location of the industrial areas near some important residential places. Mean $NO_2$ concentration measured at the industrial sites of Abidjan (20.9 ppb) is of the same order of magnitude than concentrations measured at the urban site of Singapore (23.8-28.1 ppb) in South East Asia reported by (He et al., 2014). It remains lower than that obtained

at the traffic sites of Dakar (31.7 ppb) and Al Ain city in the Middle East (31.5 ppb) (Adon et al. (2016) and Salem et al. (2009)). The levels of $NO_2$ concentration observed at Abidjan's traffic, residential and domestic fires sites are quite comparable to those measured at Bamako's traffic site (16.2 ppb).

The $NH_3$ concentration is the highest measured on all the sites presented in Table 3. The mean concentration of $NH_3$ (84.9 ppb) measured at the domestic fires sites in Abidjan is almost twice as high as that obtained in Bamako (46.7 ppb) and more

than 30 ppb higher than that obtained in Cairo (50.9 ppb) (Hassan et al., 2013). The average concentration of $NH_3$ obtained on the traffic sites of Abidjan (27.2 ppb) is also higher than that of all the other traffic sites (except for Bamako and Cairo).

In Abidjan, $HNO_3$ concentrations present a low variability for all the different sites. The highest concentration measured at the waste burning site with 1.3 ppb is equal to that obtained at the traffic sites of Dakar (1.3 ppb) and above all the concentrations of $HNO_3$ presented in Table 3.

The average concentration of $SO_2$ measured at the traffic sites in Abidjan (4.2 ppb) is 2 to 4 times higher than the concentrations obtained at the other sites in Abidjan and close to concentrations measured at the Bamako's traffic site (3.6 ppb) and on the traffic site of Al Ain city (5.8 ppb). However, $SO_2$ concentration in Abidjan remains much lower than that obtained in Dakar (15.9 ppb), Cairo (13 ppb) and Singapore (12.5-14.9 ppb). It highlights significant differences in the sources of gaseous pollutants and their level of concentrations in West African and African cities and those in developing countries.

The most important ozone concentrations in Abidjan are obtained for the waste burning (17.6 ppb) and suburban sites (12.2 ppb). The ozone concentration is naturally higher than those obtained at the traffic sites of Abidjan, Dakar and Bamako. However, at the suburban site of Amersfoort, which is highly impacted by anthropogenic activities, higher concentrations than those of Abidjan were observed. For example, 23 ppb were measured over 2007-2008 (Lourens et al., 2011) and 27 ppb over the period 1995-2005 (Martins et al., 2007). The ozone concentrations in the African cities are much lower than those measured

in the industrialized city of Shenzhen in China (35.6 ppb) (Xia et al., 2016).

### 3.3 Overview of urban $NO_2$ and $SO_2$ monitoring studies in African cities

Air quality monitoring networks are very important in assessing the health risk associated with exposure to gaseous pollutants. They also help governments to take measures of mitigation of the levels of pollutant concentrations and to define air quality indices (AQI) adapted to each country. If some synthesis of air quality measurements for particulate matter concentrations

exist in the literature at the scale of Africa (Petkova et al., 2013), there is no overview for gaseous pollutants. Nowadays, it is quite difficult to provide an overview of air quality results for gaseous pollution over Africa because of the availability of only some non-perennial studies carried out within the framework of scientific research programs. The available results are also not always communicated publicly or in the literature. As this study shows, one of the main gaseous sources of air pollution in African cities is traffic road emitting $NO_2$ and $SO_2$. This section aims to report the main studies on $NO_2$ and $SO_2$ conducted at

different time scale (hourly, daily, weekly, monthly and annually) in African urban and industrial environments. This is the result of literature searches based on studies conducted in Africa between 2005 and 2017. Publications in which methodology or data reporting were clearly describe were selected. Recent and unpublished studies have also been presented for comparison.



Figure 12 presents the African countries, coloured by UN sub-region, for which $NO_2$ and $SO_2$ monitoring studies were identified and included in this work. The different sub-regions have different colours: red for the Northern Africa, yellow for Western Africa, orange for Eastern Africa, green for Central Africa and purple for Southern Africa. We found 22 publications on $NO_2$ and $SO_2$ measurements conducted at least in 20 African countries out of 54. We found that most of those measurements

were located in seven West African countries (Senegal, Mali, Burkina Faso, Cote d'Ivoire, Nigeria, Ghana and Benin) and in five East African countries (Sudan, Kenya, Ethiopia, Uganda and Tanzania). Studies were also identified for four countries (Algeria, Tunisia, Morocco and Egypt) in Northern Africa and three countries (South Africa, Angola and Mozambique) in Southern Africa. Only one study was identified in Central Africa (Cameroon). To compare the levels of gaseous concentration in Africa with WHO air quality guidelines, we present the concentration level of each pollutant for each country using the

mean, the minimum and the maximum values when available. Figure 13 and Figure 14 represent $NO_2$ and $SO_2$ levels in African cities as reported in the various studies respectively. Green bars represent lower and upper range of means if reported. Black points represent average concentration of gaseous pollutants. Red vertical lines illustrate current WHO guidelines at different time scale. Daily or hourly routine $NO_2$ and $SO_2$ do not exist in most of the African cities. Thus, only few studies allow the comparison with WHO standards. In most of the studies we found, monitoring campaigns were carried out with different

methods. Some studies have used passive samplers of gases and others used real time active analysers or measurements using satellite. This makes difficult to compare results. The majority of these studies were carried out over fairly short periods.

*Nitrogen dioxide ($NO_2$)*

According to $NO_2$, short time exposure (1hour) at a concentration of 200 µg.m$^{-3}$ is toxic and has significant health effects. The three studies at this time scale realized by Jackson, (2005) in Tanzania and Adon et al. (2016) in Senegal and Mali show that

hourly $NO_2$ concentration range between (18 µg.m$^{-3}$ - 53 µg.m$^{-3}$), (9.4 µg.m$^{-3}$ - 150 µg.m$^{-3}$) and (15.8 µg.m$^{-3}$ - 135 µg.m$^{-3}$) in Dar-es-Salam, Dakar and Bamako respectively (Fig. 13). These levels of concentration are under the WHO threshold. WHO air quality guidelines value for annual exposure to $NO_2$ concentration is 40 µg.m$^{-3}$. Seven studies were carried during at least one year in Dakar (Senegal), Ouagadougou (Burkina Faso), Bamako (Mali), Abidjan (Cote d'Ivoire), South Africa and Ben Arous (Tunisia) (Bahino PhD, 2017 pers. com; Chaaban, 2008; Claire Demay et al., 2011; Josipovic et al., 2010; Nanaa et al.,

2012). The average annual concentration of $NO_2$ in Ben Arous, Tunisia (178 µg.m$^{-3}$) and in Dakar, Senegal (59.6 µg.m$^{-3}$) are upper the WHO threshold value when the annual average concentration measured in Dakar, Ouagadougou and in South Africa range between (3.5 µg.m$^{-3}$ and 19.6 µg.m$^{-3}$), (22 µg.m$^{-3}$ and 27 µg.m$^{-3}$) and (2 µg.m$^{-3}$ and 7 µg.m$^{-3}$) respectively. Almost nineteen studies were conducted at daily, weekly and monthly scales. 8 studies were conducted on a daily basis. Average 24-hour concentrations ranged from 0.1 µg.m$^{-3}$ along Juja road in Nairobi (Kaboro, 2006) to 752 µg.m$^{-3}$ on the traffic site of

Dantokpa market in Cotonou (Mama et al., 2013). Only 3 studies measuring $NO_2$ on a weekly scale were identified. The maximum concentration (1015.34 µg.m$^{-3}$) was measured in the city of Oweri, the capital of the oil-rich region of Imo state in Nigeria (Ibe et al., 2016). Except this extreme value, concentrations range from 9.32 µg.m$^{-3}$ in urban sites of Kampala (Uganda) to 60 µg.m$^{-3}$ in low-income areas of Accra (Ghana). Average monthly concentration is ranged from 2.6 µg.m$^{-3}$ to 87.2 µg.m$^{-3}$ for 8 studied sites. The lowest values were measured in Maputo (Mozambique) on a residential site and on a green area. The

monthly average concentration in Maputo is 9.13 µg.m$^{-3}$ and varies between 2.6 µg.m$^{-3}$ and 21 µg.m$^{-3}$ (Cumbane et al., 2008). The highest monthly average concentration of $NO_2$ was measured in Bamako. Concentrations measured on traffic and industrial sites in Abidjan (33.5 µg.m$^{-3}$ and 39.3 µg.m$^{-3}$ respectively) in this study are of the same order of magnitude as the values obtained by (Liousse and Galy-Lacaux, 2010) at urban sites in Abidjan (29.8 µg.m$^{-3}$) and Yaoundé (33.3 µg.m$^{-3}$). However, $NO_2$ measurements in Abidjan at hourly and annual scale that allow comparisons with WHO thresholds are not available or

not published. It would therefore be wise for public authorities in Abidjan as well as in other African cities to carry out regular studies at comparable time scales and with recommended methods to obtain reliable results.





### Sulfur dioxide (SO$_2$)

WHO indicates that changes in pulmonary function and respiratory symptoms appear after a ten-minute exposure period to SO$_2$ at a concentration of 500 µg.m$^{-3}$. No study carried out with a minimum time interval of 10 minutes was reported in the literature. Twenty-four hour SO$_2$ levels is significantly associated with daily mortality rates (Burnett et al., 2004). WHO

twenty-four hour guideline for SO$_2$ is 20 µg.m$^{-3}$. Nine daily (24-hour) studies are reported on Fig. 14. Measurements in Cotonou (Benin) at the crossroad of Dantokpa market are characterized by a large circulation of two-wheeled vehicles and daily average concentrations range between 784.8 µg.m$^{-3}$ and 3662.4 µg.m$^{-3}$ (Mama et al., 2013). These concentrations are 39 to 183 times higher than the threshold value. In Dakar (68.54 µg.m$^{-3}$), Cairo (34 µg.m$^{-3}$), and Bamako (29.03 µg.m$^{-3}$), SO$_2$ mean daily concentrations are greater than 20 µg.m$^{-3}$. It is found that only 2 studies conducted in Marrakech (Morocco) and Tunis (Tunisia)

have concentration ranges lower than the threshold. In Nairobi and Khartoum the maximum concentration can reach and exceed the threshold.

Hourly SO$_2$ concentration in Africa is found to be ranged from 9.14 µg.m$^{-3}$ to 214514 µg.m$^{-3}$. In Luanda (Angola), concentrations are the highest and range between 13080 µg.m$^{-3}$ and 214514 µg.m$^{-3}$ (Amadio, 2010). We must note that this study focused on SO$_2$ content in diesel engine releases. In West Africa (Mali and Burkina), concentrations are much lower (9

µg.m$^{-3}$ - 136 µg.m$^{-3}$).

Approximately 6 studies report weekly and monthly concentrations of SO$_2$ (Fig. 14). Generally, weekly and monthly SO$_2$ concentrations in Africa range from 0.77 µg.m$^{-3}$ to 18.9 µg.m$^{-3}$. The weekly concentrations measured in the oil state of Imo State (Nigeria) are not in this range (1203 µg.m$^{-3}$ to 1465 µg.m$^{-3}$) (Ibe et al., 2016).The annual average concentrations of SO$_2$ are reported for 11 studies. The concentrations measured in North Africa are the highest. Average levels of SO$_2$ range between

8 µg.m$^{-3}$ in Rabat (Morocco) and 325 µg.m$^{-3}$ in Alger (Algeria). In West Africa, concentrations are lower and range from 0.5 µg.m$^{-3}$ in Ouagadougou (Burkina Faso) to 80 µg.m$^{-3}$ in Dakar (Senegal). The annual average SO$_2$ concentration in Abidjan (3.66 µg.m$^{-3}$) is one of the lowest of West African cities reported in this study (Bahino PhD, 2017 pers.com). Measurements of SO$_2$ at smaller time scales (10 min, 24 hours…) do not exist in Abidjan or are not published. As for NO$_2$, studies should be carried out regularly and according to WHO standards.

### 4 Conclusion and recommendations

Abidjan is the economic capital of the West African sub region. It is a large industrial and commercial city. Abidjan has also the second most important port in sub-Saharan Africa. The new urban master plan of the city called "The Great Abidjan" foresees its extension to 6 peripheral municipalities. The district of Abidjan will increase from 13 to 19 municipalities by the year 2030. This extension will be accompanied by major infrastructure works which will include the construction of 7 new

bridges, new access roads and bypasses to the city center, the creation of 2 metro lines, the opening of new industrial and commercial areas… In total, nearly hundred projects will be realized. Projections estimate a demographic growth that will increase the population of nearly 5,000,000 inhabitants to 8,414,000. The city will thus be more subject to the demographic pressure, the uncontrolled occupation of the land and the emission of gaseous and particulate pollutants.

The present study reports the gaseous concentrations measurements using passive samplers performed during an intensive

experimental field campaigns in the framework of the Work Package 2 (Air Pollution and Health) of the European DACCIWA project. This work presents an original database of bi-monthly gaseous concentrations (NO$_2$, SO$_2$, NH$_3$, HNO$_3$, O$_3$) in the district of Abidjan performed on 21 representative sites. This database allows for the first time to characterize the levels of gaseous concentration and to present a spatial distribution of gaseous pollution at the scale of the city of Abidjan during the dry season (December 15$^{th}$, 2015 to February 16$^{th}$, 2016).

Our results show that there is a great spatial variability of gaseous pollutants in Abidjan. The average concentrations of the main gaseous pollutants are ranged between (2.7 ppb - 25 ppb) for NO$_2$, (9.1 ppb - 102.1 ppb) for NH$_3$, (0.2 ppb - 1.4 ppb) for



HNO₃, (0.4 ppb - 7.21 ppb) for SO₂ and (5.1 ppb - 19.11 ppb) for O₃. The spatial distribution of gaseous pollutant have been studied according the main potential sources of pollution nearby the measurement sites. Results show that the concentration level of gaseous pollutants such as NH₃ and NO₂ is very important at some sites. On one hand, it highlights the predominance of the domestic fires source with the use of wood and charcoal that emits large amount of NH₃. On the second hand, it shows

the significant impact of traffic road on NO₂ and SO₂ emissions. The car fleet is aging and restrictive measures on the importation of these vehicles are not always respected. Public authorities should introduce control of pollutant emissions at the exhaust of vehicles and improve road fluidity in order to reduce NO₂ and SO₂ emission.

By grouping the sites and calculating the average gases concentrations per source of pollution, we show that industrial (20.9 ppb), domestic fires (17.9 ppb) and traffic (17.8 ppb) sites largely contribute to the NO₂ pollution. Residential sites with an

NO₂ average concentration of 16.1 ppb are strongly influenced by road traffic. Concerning NH₃, domestic fires sites constitute by far the first sources of pollution with an average concentration of 84.9 ppb, almost two times higher than in Bamako and Cairo. The waste burning site characterized by the continuous combustion of different kinds of waste is the second main source of pollution by NH₃ (39.1 ppb). HNO₃ concentrations remain in a low range between 0.6 ppb and 0.9 ppb, except for the waste burning site (1.3 ppb). Abidjan, the other African cities and the big cities of Asia have almost identical average concentration

of HNO₃. In addition to the high concentration of NO₂, the traffic sites have the highest SO₂ average concentration (4.2 ppb). NO₂ concentration measured on Abidjan industrial and traffic sites are generally of the same order of magnitude as those measured in the most of the big cities of Asia and Africa. SO₂ average concentration on the other sites is between 1 and 2 ppb. O₃ concentrations seem to be more or less constant over the city sites with an average concentration varying between 9 ppb and 10 ppb. Waste burning (17.6 ppb) and suburban (12.2 ppb) sites have the highest O₃ average concentration.

In this study, we built for the first time an overview of gaseous SO₂ and NO₂ pollutants which allows to give a global picture of gases concentrations at the scale of several African cities. It includes 22 publications on NO₂ and SO₂ measurements conducted in 20 African countries during the period 2005-2017. This study also emphasizes that in Abidjan, as in most of the major African cities, there are no air quality monitoring networks. Only a few studies carried out by research teams within the framework of scientific programs with various objectives give an idea of the state of air quality. These studies used different

methods and time scales that do not allow comparison with WHO standards and threshold values. The comparable studies show that the hourly concentration of NO₂ in African cities (9.4 µg.m⁻³ and 135 µg.m⁻³) is lower than the WHO standards. Abidjan annual NO₂ concentration is 23.25 µg.m⁻³ with a range at the African cities scale from 2 µg.m⁻³ to 175 µg.m⁻³. Annual NO₂ WHO standards are exceeded in Dakar and Tunis. The daily concentration of SO₂ in African cities ranges from 0.2 µg.m⁻³ to 3662 µg.m⁻³ and exceeds the daily SO₂ WHO standard (40 µg.m⁻³) in most of the African cities, except in Marrakech and

Tunis. These scarce datasets should draw the attention of the African public authorities to the urgent need to measure NO₂ and SO₂ emissions.

The results indicate the need of other studies including VOCs and NOx measurements to better understand the production processes of secondary pollutants such as O₃. In the framework of DACCIWA WP2 (Air Pollution and Health) program, bi monthly measurements are ongoing since December 2014 over three supersites. These measurements will make possible to

estimate the impact of pollutants concentrations on the population health through the calculation of real dose response functions.

In Abidjan as well as in other African cities, legislations are taken to define national standards. Recent initiatives of the State of Cote d'Ivoire to limit and control emissions of pollutants into the atmosphere let hope for an improvement of air quality. Nowadays, infrastructures under construction coordinated by the PRICI project (Projet de Renaissance des Infrastructures de

Cote d'Ivoire) and financed by the AFrican Development Bank (AFDB) are part of sustainable development. The government of Cote d'Ivoire has also adopted a decree (no. 2017-125) on air quality in the Council of Ministers on 22 February 2017. On





the one hand, the decree fixes the limit values for the ambient air gases and particulates concentrations emitted by motor vehicles and motorcycles. On the other hand, it defines air quality control procedures as well as the potential penalties. The authorities should continue their efforts by establishing a real air quality monitoring network in agreement with international standards.

5  **Acknowledgment**

This work has received funding from the European Union 7th Framework Programme (FP7/2007-2013) under Grant agreement no. 603502 (EU project DACCIWA: Dynamics-aerosol-chemistry-cloud interactions in West Africa). The authors wish also to thank the AMRUGE-CI (Appui à la Modernisation et à la Réforme des Universités et Grandes Ecoles de Cote d'Ivoire) project.



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





**List of Tables**





**Table 1: Geographical coordinates of sampling sites and majors source of air pollution nearby.**

| ID code° | Site Name | Municipality | Latitude | Longitude | Land Use |
|---|---|---|---|---|---|
| A1 | 220 Lgts Liberté | Adjame | 5°21'14″ N | 4°01'04″ W | Traffic |
| A2 | Gesco | Yopougon | 5°21'31″ N | 4°06'03″ W | Traffic |
| A3 | N'Dotré | Abobo | 5°26'41″ N | 4°04'10″ W | Traffic |
| A4 | Corridor Anyama | Anyama | 5°31'8 ″ N | 4°03'34″ W | Suburban |
| A5 | Pharmacy Cadre Blvd | Plateau | 5°19'33″ N | 4°01'26″ W | Traffic/ Administrative center |
| A6 | Airport-FHB | Port-Bouet | 5°16'26″ N | 3°55'21″ W | Traffic |
| A7 | Town hall Attecoube | Attecoube | 5°19'52″ N | 4°02'23″ W | Popular district / Traffic |
| A8 | Town hall Abobo | Abobo | 5°25'15″ N | 4°01'00″ W | Traffic |
| A9 | Yopougon industrial area | Yopougon | 5°22'12″ N | 4°04'52″ W | Industrial |
| A10 | Zone 3 | Marcory | 5°17'48″ N | 3°59'57 »W | Industrial |
| A11 | Tri postal-vridi | Port-Bouet | 5°16'12″ N | 4°00'07″ W | Industrial / harbor |
| A12 | University FHB | Cocody | 5°20'42″ N | 3°59'27″ W | Residential district |
| A13 | Angré | Cocody | 5°23'27″ N | 3°59'34″ W | Residential district |
| A14 | Place Inch'allah | Koumassi | 5°17'52″ N | 3°57'20″ W | Domestic fires/Residential |
| A15 | Akouédo | Cocody | 5°21'12″ N | 3°56'16″ W | Waste burning |
| A16 | Niangon Bracodi | Yopougon | 5°19'44″ N | 4°06'21″ W | Domestic fires |
| A17 | Scientific pole CNRI-UFHB | Bingerville | 5°21'30″ N | 3°54'07″ W | Suburban background |
| A18 | Songon health center | Songon | 5°19'6″ N | 4°12'06″ W | Suburban background |
| A19 | Niangon Adjamé | Yopougon | 5°20'15″ N | 4°07'08″ W | Suburban background |
| A20 | Gonzagueville | Port-Bouet | 5°14'31″ N | 3°53'09″ W | Suburban |
| A21 | Ecological research center | Treichville | 5°18'41″ N | 4°00'10″ W | Green area |





**Table 2: Gaseous pollutants concentrations and standard deviation on 21 sampling sites in Abidjan during the dry season (December 15th, 2015 - February 16th, 2016).**

| \multicolumn Sampling sites | | | Concentration in ppb | | | | |
|---|---|---|---|---|---|---|---|
| ID Code | Name | Land use | $NO_2$ | $NH_3$ | $HNO_3$ | $SO_2$ | $O_3$ |
| A1 | 220 Lgts Liberté | Tr | 20.8±5.0 | 31.1±6.1 | 1.1±0.1 | 2.6±1.4 | 11.3±1.0 |
| A2 | Gesco | Tr | 11.6±1.0 | 27.6±2.3 | 0.6±0.1 | 6.8±1.3 | 12.4±1.0 |
| A3 | N'Dotré | Tr | 12.2±1.8 | 23.5±2.4 | 0.4±0.1 | 7.2±1.2 | 8.9±0.9 |
| A4 | Corridor Anyama | Tr/Df | 7.7±0.6 | 32.2±3.8 | 0.2±0.0 | 4.7±0.8 | 5.1±0.6 |
| A5 | Pharmacy Cadre Blvd | Tr | 23.9±4.9 | 23.0±2.2 | 1.1±0.4 | 3.1±1.1 | 7.8±0.4 |
| A6 | Airport-FHB | Tr | 6.9±0.9 | 18.8±1.9 | 0.9±0.1 | 0.4±0.2 | 18.8±3.0 |
| A7 | Town hall Attécoubé | Pop/Tr | 17.5±2.9 | 37.0±10.9 | 1.2±0.5 | 1.5±0.5 | 12.4±2.0 |
| A8 | Town hall Abobo | Pop/Tr | 20.4±2.1 | 30.6±3.6 | 1.1±0.3 | 1.4±0.6 | 10.5±1.9 |
| A9 | Yopougon industrial area | Ind | 19.9±3.4 | 21.6±2.8 | 1.3±0.2 | 1.8±0.7 | 12.2±1.3 |
| A10 | Zone 3 | Ind/Tr | 17.7±1.0 | 20.9±2.9 | 0.8±0.1 | 1.6±0.4 | 8.8±1.5 |
| A11 | Tri postal-vridi | Ind/Har | 25.0±1.7 | 19.6±1.6 | 0.7±0.2 | 1.9±0.5 | 9.6±1.9 |
| A12 | University FHB | Res | 12.5±2.4 | 23.9±5.0 | 1.0±0.4 | 0.7±0.1 | 14.1±0.4 |
| A13 | Angré | Res | 18.5±2.2 | 26.9±2.4 | 1.2±0.3 | 1.7±0.1 | 12.9±1.9 |
| A14 | Place Inch'allah | Df/Tr | 17.6±1.9 | 67.7±8.3 | 0.6±0.1 | 1.1±0.3 | 7.6±2.4 |
| A15 | Akouédo | Wb | 12.0±1.4 | 39.1±2.1 | 1.3±0.2 | 1.2±0.7 | 17.6±1.9 |
| A16 | Niangon Bracodi | Df | 18.3±0.6 | 102,1±9.1 | 1.2±0.2 | 1.1±0.2 | 10.2±1.4 |
| A17 | Scientific pole cnri-UFHB | Sub | 13.6±3.7 | 22.5±6.1 | 1.4±0.3 | 1.0±0.3 | 16.8±3.0 |
| A18 | Songon Heath center | Sub | 2.7±0.1 | 9.1±1.7 | 0.7±0.2 | 1.9±0.2 | 12.1±1.3 |
| A19 | Niangon Adjamé | Sub | 5.4±0.3 | 44.0±2.2 | 0.2±0.0 | 3.4±1.0 | 6.8±0.7 |
| A20 | Gonzagueville | Sub/Df | 6.9±0.5 | 14.3±2.6 | 0.8±0.4 | 0.4±0.1 | 19.1±1.7 |
| A21 | Ecological research center | Gre | 6.0±0.7 | 13.6±2.9 | 0.6±0.2 | 0.4±0.1 | 13.6±1.4 |



**Table 3: gaseous pollutants concentrations (ppb) in different sites of Abidjan and in other cities of developing countries**

| Sites | Type | Period | NO₂(ppb) | NH₃(ppb) | HNO₃(ppb) | SO₂ (ppb) | O₃(ppb) | References |
|---|---|---|---|---|---|---|---|---|
| Abidjan, Cote d'Ivoire | Traffic | Dec. 2015-Feb. 2016 | 17.8 | 27.2 | 0.9 | 4.2 | 10.2 | This study |
| | Industrial | Dec. 2015-Feb. 2016 | 20.9 | 20.7 | 0.9 | 1.7 | 10.2 | This study |
| | Residential | Dec. 2015-Feb. 2016 | 16.1 | 21.9 | 0.8 | 1.0 | 9.9 | This study |
| | Suburban | Dec. 2015-Feb. 2016 | 7.0 | 22.6 | 0.6 | 2.0 | 12.2 | This study |
| | Domestic fires | Dec. 2015-Feb. 2016 | 17.9 | 84.9 | 0.9 | 1.1 | 8.9 | This study |
| | Waste burning | Dec. 2015-Feb. 2016 | 12.0 | 39.1 | 1.3 | 1.2 | 17.6 | This study |
| Dakar, Senegal | Traffic | Jan. 2008-Dec.2009 | 31.7 | 21.1 | 1.3 | 15.9 | 7.7 | (Adon et al., 2016) |
| Bamako, Mali | Traffic | Jun. 2008 - Dec. 2009 | 16.2 | 46.7 | 0.6 | 3.6 | 5.1 | (Adon et al., 2016) |
| Cairo, Egypt | Suburban | Winter of 2009-2010 | 35.5 | 50.9 | - | 13.0 | - | (Hassan et al., 2013) |
| Amersfoort, SA* | Suburban, Industrialized | Aug. 2007-Jul. 2008 | 2.9 | - | - | 5.5 | 23.3 | (Lourens et al., 2011) |
| Amersfoort, SA* | Suburban, Industrialized | 1995-2005 | 2.5 | 1.2 | 0.9 | 2.8 | 27 | (Martins et al., 2007) |
| Singapore | Urban | Sept. 2007- Aug. 2008 | 23.8-28.1 | - | - | 12.5-14.9 | - | (He et al., 2014) |
| Al Ain City, UAE | Traffic | Feb.2005-Feb.2006 | 31.5 | 17.1 | - | 5.8 | 8.7 | (Salem et al., 2009) |
| Shenzhen, China | Urban | March. 2013-Feb.2014 | 20.7 | - | - | 4.9 | 35.6 | (Xia et al., 2016) |
| St. John's. India | Urban | | 4.8 | 9.5 | 0.86 | 1.56 | - | (Kumar et al., 2004) |

*SA: South Africa





## List of Figures





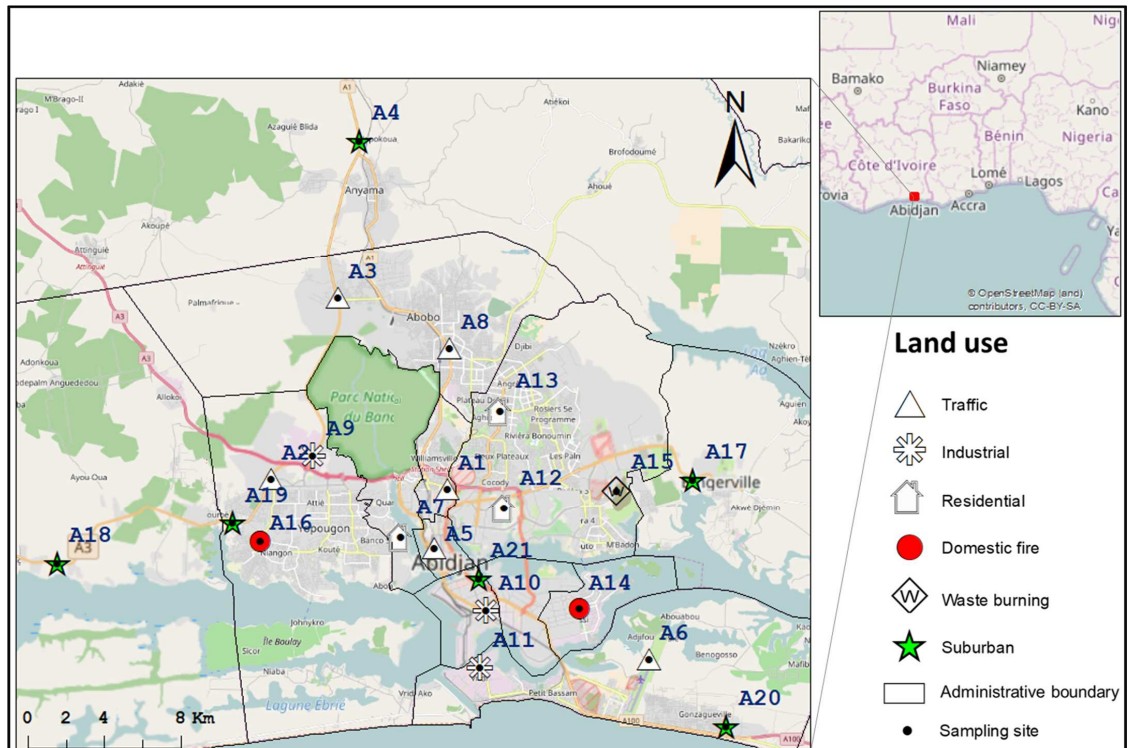

Figure 1: Spatial distribution of the 21 sampling sites (A₁-A₂₁) and major source of air pollution nearby in the District of Abidjan.



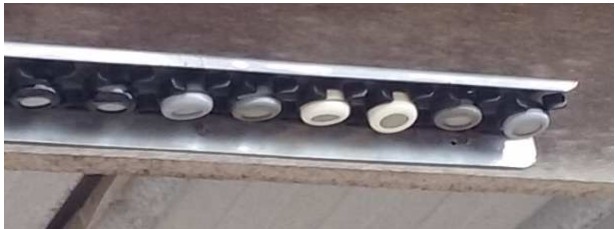

**Figure 2: INDAAF passive samplers mounted in duplicate on a stainless steel holder.**





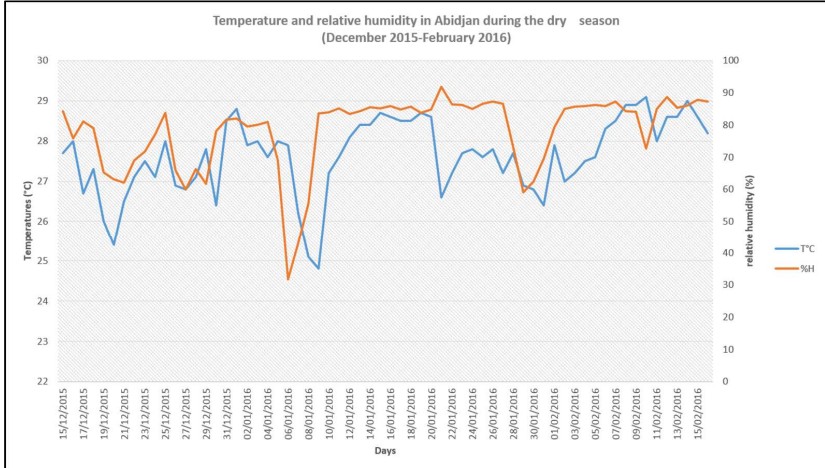

**Figure 3: Temperature and relative humidity measured by (ASECNA) in Abidjan during the dry season (December 15th, 2015 – February 16th, 2016).**





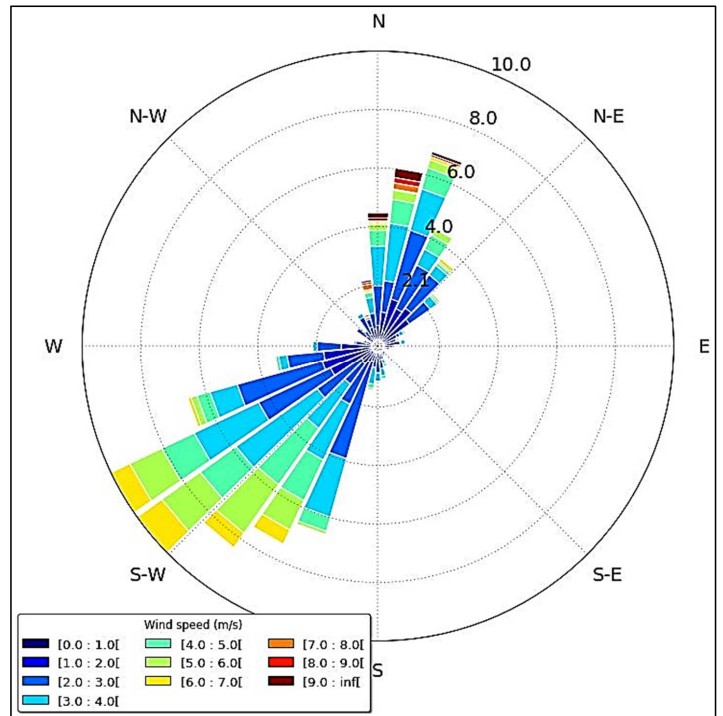

**Figure 4: Hourly wind rose diagram of Abidjan during the dry season (December 15th, 2015 – February 16th, 2016)**

.





**Figure 5: Regression between interpolated data and observed data using kriging method. Fig 5.a for NO₂, SO₂, HNO₃, O₃. Fig 5.b for NH₃ determined by removing 2 extremes values)**





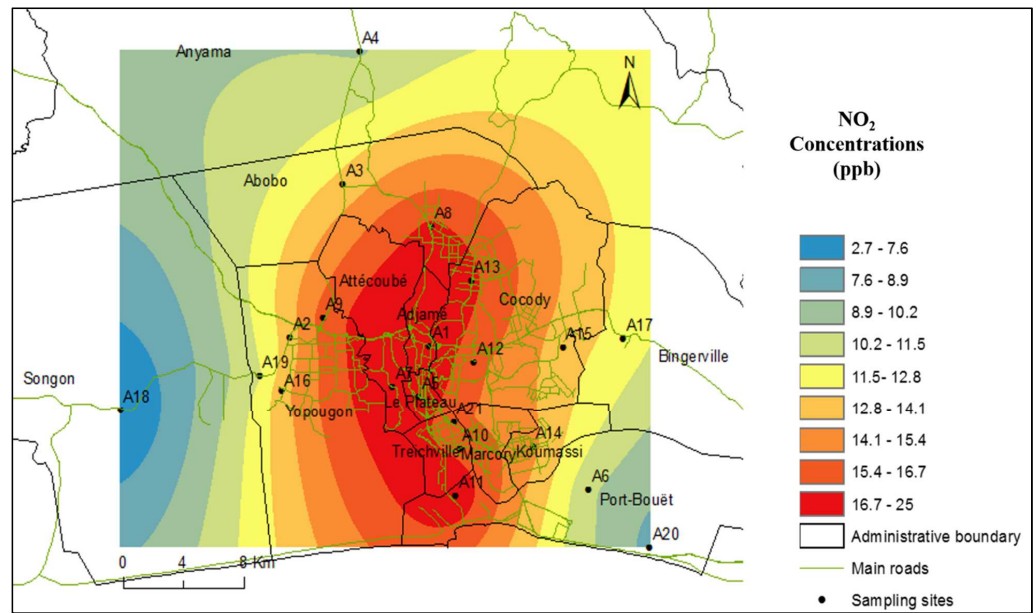

**Figure 6: Spatial distribution of NO₂ in the district of Abidjan during the dry season (December 15th, 2015 – February 16th, 2016).**





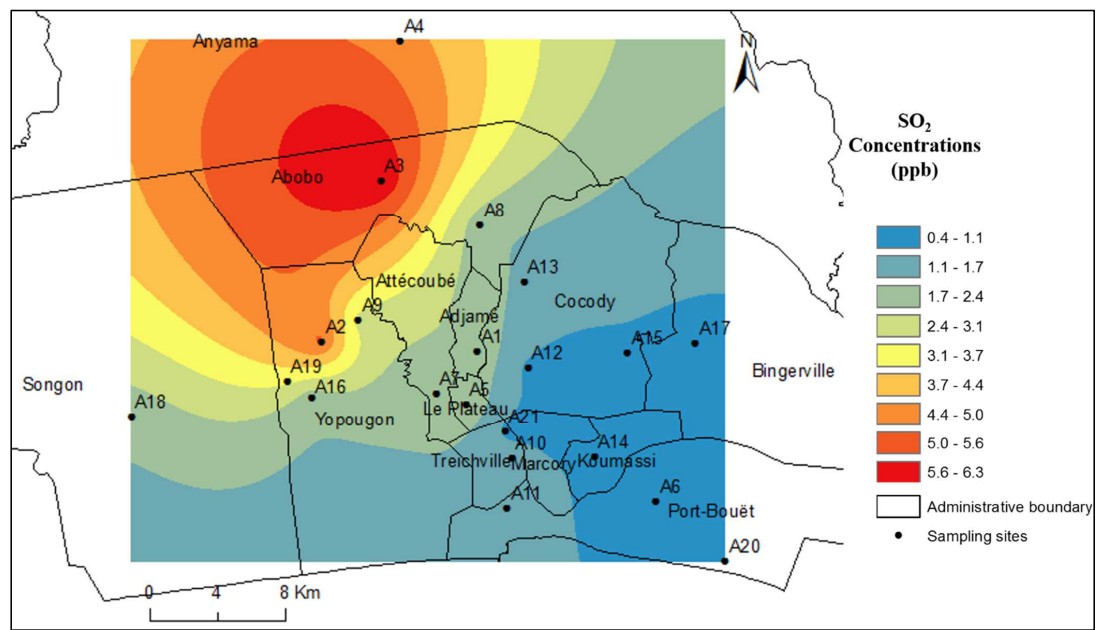

**Figure 7: Spatial distribution of SO₂ in the district of Abidjan during the dry season (December 15ᵗʰ, 2015 – February 16ᵗʰ, 2016).**





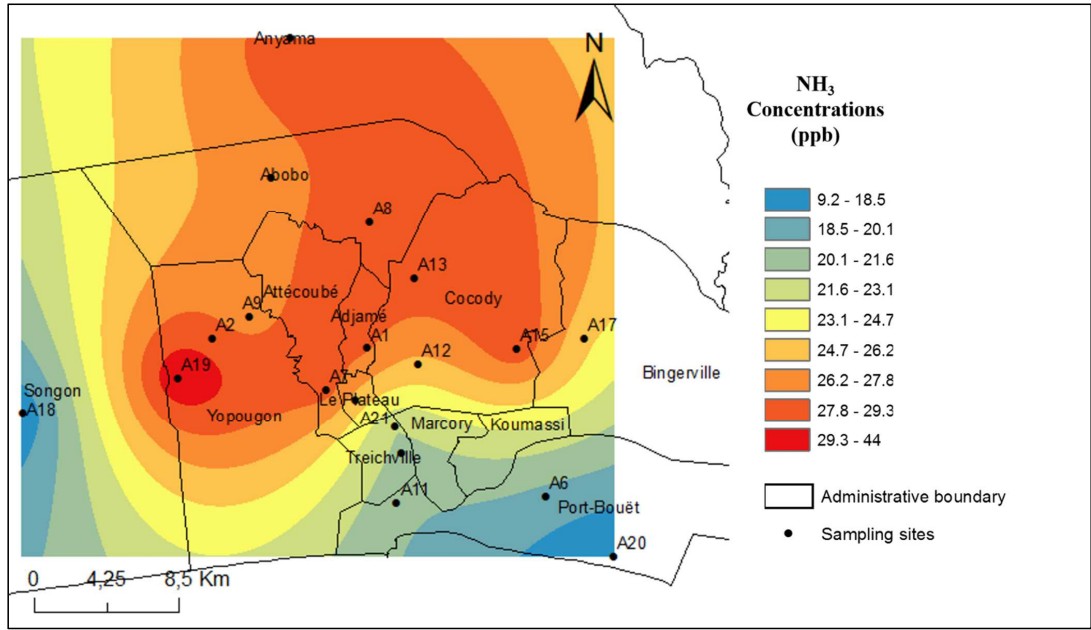

**Figure 8: Spatial distribution of NH₃ in the district of Abidjan during the dry season (December 15th, 2015 – February 16th, 2016).**





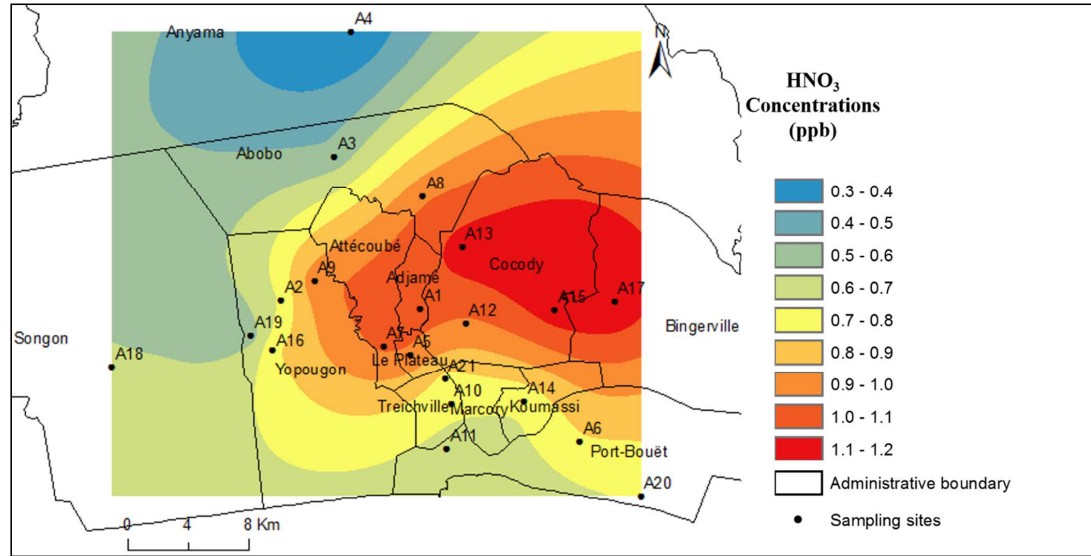

**Figure 9: Spatial distribution of HNO₃ in the district of Abidjan during the dry season (December 15ᵗʰ, 2015 – February 16ᵗʰ, 2016).**



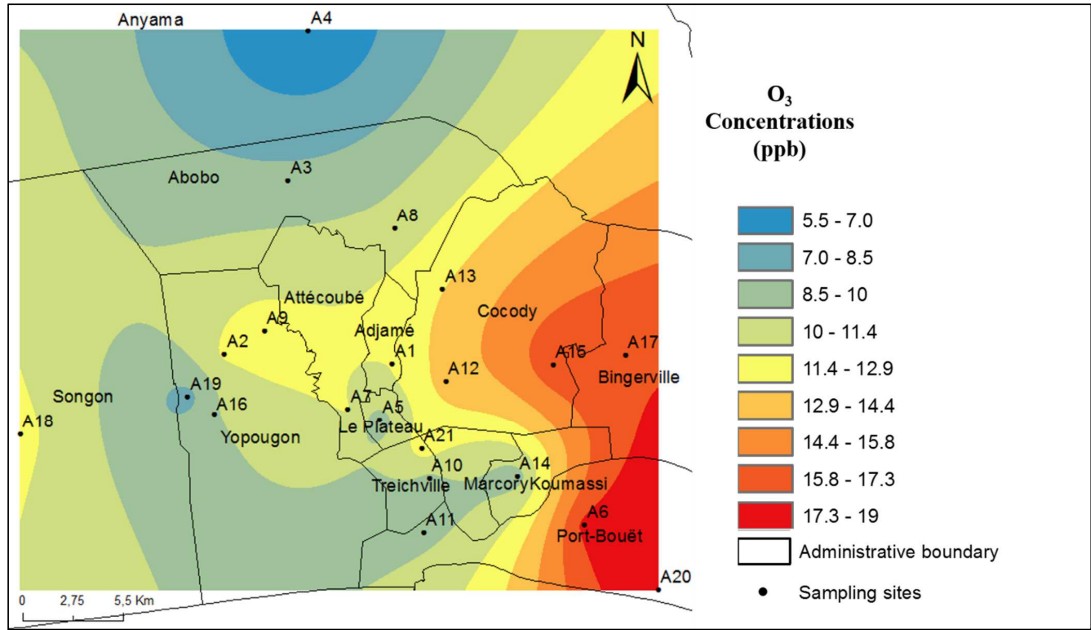

**Figure 10: Spatial distribution of O₃ in the district of Abidjan during the dry season (December 15th, 2015 – February 16th, 2016).**



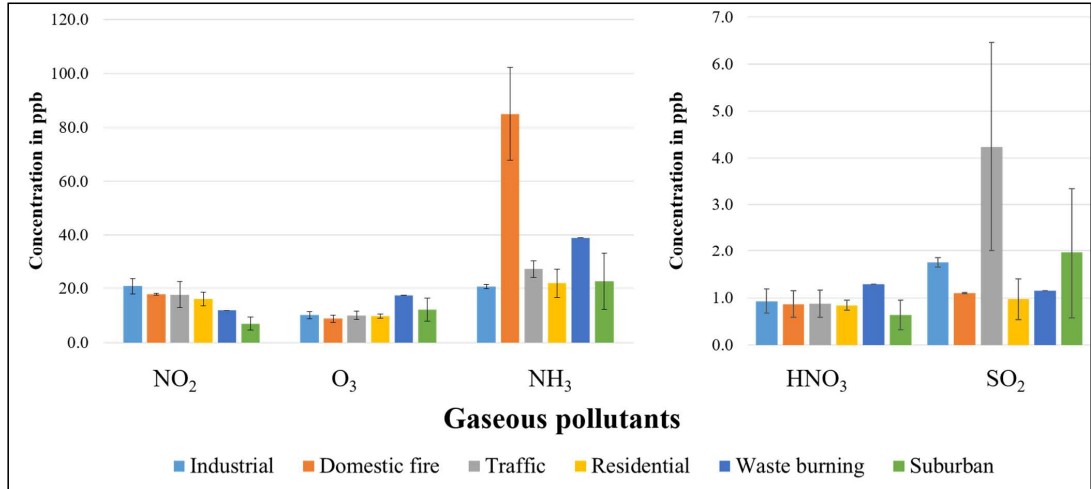

**Figure 11: Concentration of gaseous pollutants by anthropogenic sectors of activity in the district of Abidjan.**



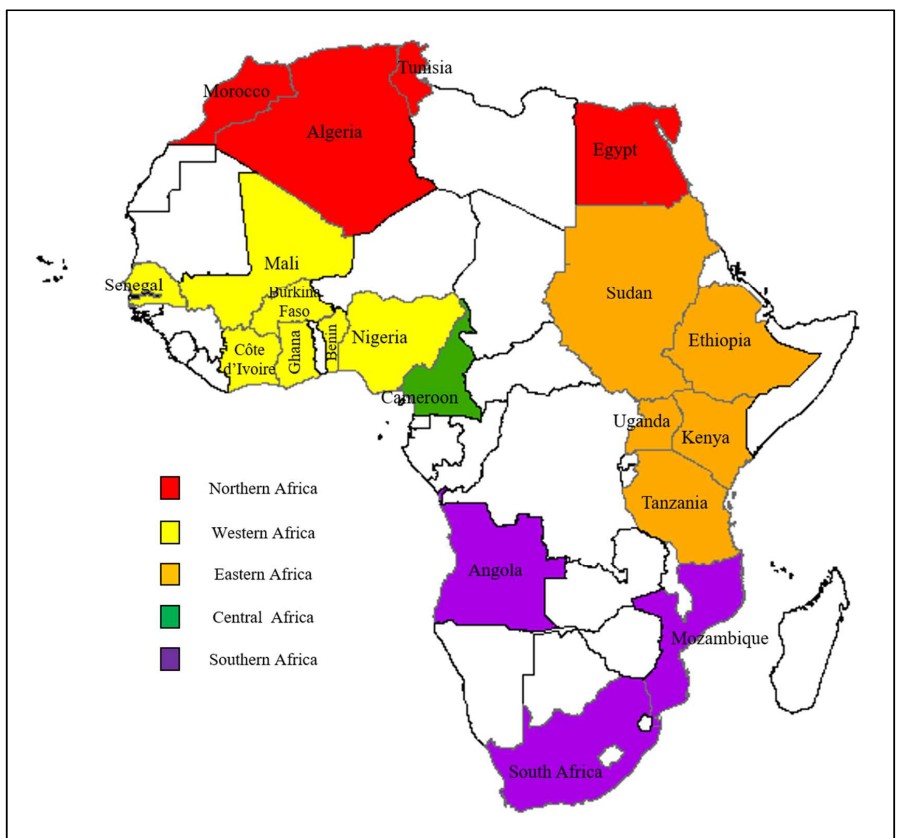

**Figure 12: African countries coloured by UN sub-region for which NO$_2$ and SO$_2$ studies were identified and included in this publication.**





**Figure 13: NO₂ concentration levels in µg.m⁻³ in African capitals at different time scale (hourly, daily, weekly, monthly, and annually) as reported in literature. Green bars represent lower and upper range of means if reported. Black points represent average concentration of NO₂. Red vertical lines illustrate current WHO guidelines at different time scale.**





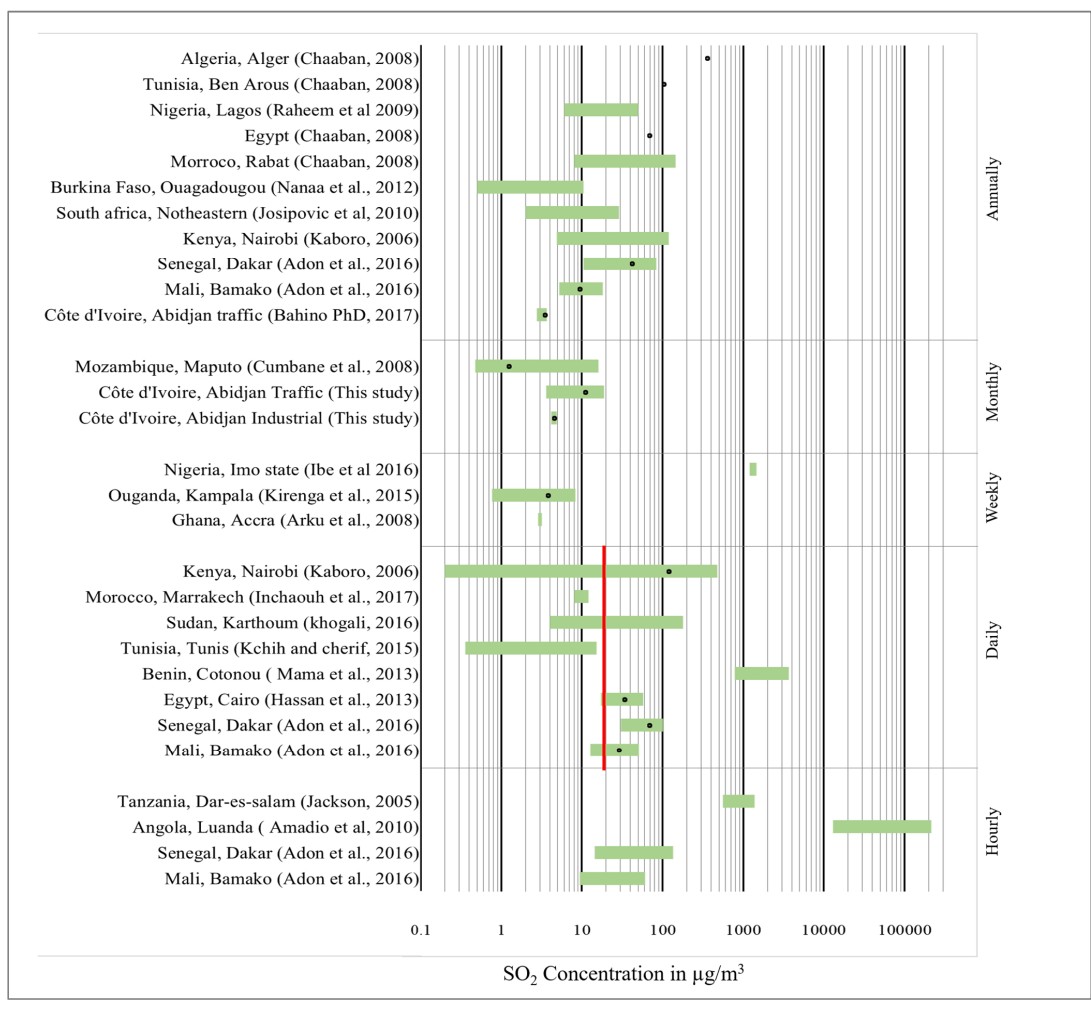

5   **Figure 14: SO₂ concentration levels in µg.m⁻³ in African capitals at different time scale (hourly, daily, weekly, monthly, and annually) as reported in literature. Green bars represent lower and upper range of means if reported. Black points represent average concentration of SO₂. Red vertical lines illustrate current WHO guidelines at different time scale.**