# Peer review of "A pilot study of gaseous pollutants measurement (NO2, SO2, NH3, HNO3 and O3) in Abidjan, Cote d'Ivoire: contribution to an overview of gaseous pollution in African Cities."

_Atmospheric Chemistry and Physics, 2017_

## Referee Comment (RC1) · Anonymous Referee #1 · 26 Sep 2017

This paper presents a relatively high resolution spatial distribution for inorganic gaseous species measured at Abidjan in Cote d'Ivoire. Since very little data exist for this part of Africa, I welcome any data from this region. Scientists conduct measurements under difficult conditions and results presented should be viewed within the logistical constrains associated with measurements conducted in this part of the world. However, my main concern with this paper is that results for a very short period are presented, i.e. 8 weeks during which passive sampling measurements were conducted for two-week periods. That is a very small data set, although it is considered an intensive campaign within the scope of the DACCIWA FP7 project. It would have been good if there were 8 weeks of continuous measurements, but then again the scientists would not have been able to cover the spatial distribution (21 sites). It is mentioned that one

of the 21 sites is a "supersite". However, I presume that this so-called "supersite" does not contain any continuous active measurements that could have been used to present e.g. diurnal patterns.

Therefore, my general comments on the paper to the authors are:

1) I would like to suggest that the title, introduction and structure of this paper is changed.

It should indicate that the main focus of this paper is to present and contextualise concentrations of inorganic gaseous species for African cities. Begin the paper with the spatial distributions measured for Abidjan, i.e. inorganic gases concentrations associated with specific source sectors. Thereafter the average concentrations of inorganic gases (maybe also include more than just NO2 and SO2) measured in Abidjan is compared to other African cities with the discussion and contextualization presented. Then you end of with comparing specific source sectors in Abidjan with other similar source sectors in other African countries.

To summarize, the introduction should be re-written in order to indicate that his paper deals with concentrations of inorganic gases in African cities. You have a high spatial resolution for Abidjan where you have measured and then contextualise with other African countries.

As mentioned, my main concern with this paper is the short measurement period, i.e. 8 weeks (4 samples for each gas). I do not think that it is sufficient for publication in ACP if the paper is only presented primarily from the view of a very short sampling campaign. If the paper is pitched from a perspective to review/contextualise African cities then the impact of the paper can improve.

2) The paper can be significantly shortened. In its current format, the paper is of a repetitive nature with many sentences/arguments/results repeated throughout the document. In addition, many obvious general statements are given throughout the manuscript.

Restructuring the paper will also eliminate the repetitive nature of the paper.

3) There is a lack of depth in the explanation of the observed spatial distributions. Incorporating meteorology or air mass movements, for instance, would improve the understanding and interpretations.

4) The paper also require significant language and text editing.

Detailed comments: I have included detailed specific comments relating to the above general comments in the attached PDF version of the submitted manuscript. Many language and text edits/suggestions are indicated throughout the reviewed PDF version of the manuscript.

Please also note the supplement to this comment:
https://www.atmos-chem-phys-discuss.net/acp-2017-724/acp-2017-724-RC1-supplement.pdf

[Figure]

**Supplement:**

[revised manuscript text omitted]

---

## Referee Comment (RC2) · Anonymous Referee #2 · 25 Oct 2017

Overview

It is really welcome to see a paper with new chemically detailed air quality pollutant data from Abidjan, Cote d'Ivoire. As the authors outline there is currently very little information in this region and hence the paper represents a step change in knowledge. The protocols for the passive samplers are well applied with excellent quality assurance protocols.

However in order to be sufficient to publish in ACPD, there needs to be a clearer detailing of all the data and a stronger interpretative section in the paper.

There also needs to be a detailed review of the language and word use throughout as some of the choices of words do not fit and many sentences need to be sense checked.

[Figure]

Technical and scientific comments:

1. Introduction Line 32 onwards There is a a lot of discussion of the specific project through which the research was done which is not particularly relevant for the paper. Could this be revised to be shorter and more specific to the intensive long term sites – what is being measured at them and at what resolution – if the supersite in Abidjan has been operational for the year it would be useful to know more detail about that and how the intensive reported fits in.

Beyond a primary survey of background air quality pollutant concentrations, what are the starting hypotheses and how does this study address them?

2. Experimental design P3 Line 9 onwards The economic description of Abidjan should really be in the introduction as it does not relate to the experiment. This paragraph could be shortened and refer to Figure 1 perhaps combining with section 2.1

P3 line 36: Is the Tecsult report publically available or could it be put in the supporting material? If not then rather than refer to it, the details of the report suggested network design should be in the details of this paper, and perhaps can be critiqued...did the network design meet the aims of the project?

Can you extrapolate from this number of sites to the whole region for all pollutants (my feeling is that for ozone it may be fine, but for ammonia it may not be the best approach). Referencing the global literature on kriging and air quality impact assessment would be good. A variogram fit would be useful to understand how well the kriging worked for the different pollutants. What is the topography of the area?

P4 line 4: What method was used to assign the land use type for each station?

P4 line 39: It would be useful to discuss potential interferents/artefacts: e.g. HNO3 will have interferents from other reactive oxidised N species, and NH3 potentially amines – depending on the coating type. Could the authors add a little detail on this, even though they are in the reference Adon (2010) I think it is worth re-iterating them in this

paper, perhaps in a table.

Section 2.2.2 I am impressed by the protocols used for the passive samplers. The blanks are slightly higher than I would expect for ammonia field blanks but data capture was good.

The purpose of the kriging needs to be put in context a bit more clearly: Is the purpose to derive background maps onto which high resolution data (e.g. from Supersite) can be superimposed?

Despite the high correlations, the use of emission dominated sites (e.g. the traffic sites) is likely to skew the background map high. Ideally all sites should be away from a specific strong source – however this may be what has actually done – Table 1 needs to explain the land use categories. It is noted that the advantage of measurement based concentration maps is that the avoid emission factor errors, but the drawbacks also need to be referenced.

P6: Section 3.1 It is disappointing that having had several exposure periods the authors went directly to using the average and also comparing the average of the period (2 months) to an annual average. Given that many of the pollutants have seasonal cycles as well as shorter term cycles some discussion about the appropriateness of extrapolating to an annual average is necessary.

Also it would be useful to see the data for each site for each measurement period. Perhaps a whisker plot with the average max and min or just a time series for each site and chemical?

NO2 section P7 Line 12: Comparison of short term data against the annual average is not useful. Is it expected that the NO2 is highest in the dry season or are emissions approximately constant over the year? If extrapolating to an annual average, the assumptions made should be made clear.

P7 line 24 The part about black carbon in this section does not appear to fit with the

discussion around NO2. It either needs to be removed or a discussion of black carbon correlation with NO2 from the Keita pers comm reference. If it is a pers comm, perhaps the data could be added to this paper?

Ammonia P8 line 21 Human waste (and its management) is not really mentioned as a source of ammonia though it is likely to be a significant source.

P8 line 34 The measurements <5 m from sources are correctly not used in the krigin, however for background mapping as ammonia has such a short atmospheric lifetime I would suggest that >50 m from sources is appropriate. Could the authors clarify how close the other measurement points were from NH3 sources. Again the full dataset should be presented in supplementary material so that readers can understanding the details rather than just the range being presented.

P9 line 9 at the end of the ammonia section the first part is repeated. Some tightening of the text would improve the manuscript here

Nitric acid P9 line 20: the statement that "HNO3 appears to be emitted from several sources" cannot be backed up by the measurements. HNO3 as the authors stated is frequently a secondary pollutant, rather than a primary emission. Also there are several other gas phase oxidised nitrogen chemicals which could also contribute to a passive HNO3 sampling so some discussion of potential other contributions and reference to a review or overview of the issue.

P9 line 22: I think there are papers in the literature which have made direct observations of HNO3 and therefore it would be useful to discuss the results in terms of the literature (although there is some discussion later)

Section 3.2 The discussion of the different source attributions would be improved by a more detailed description of each of the sites – perhaps a focused attribution for NO2 using smaller maps or photos. This section is useful but slightly hard to follow with just the maps and average concentrations.

Significant amounts of the text are fairly similar to the preceding sections on the measurements.

Section 3.3 The overview of the urban studies across the African continent is very useful. However it would be useful for the authors to adjust the discussion from listing the measurements and the range of values, but also to consider the impacts and purpose of the measurements. The snapshot campaign provides a picture, and it seems that Abidjan sits in a similar range to many other cities. A short atmospheric chemistry section discussing the mix of the pollutants and hence if one was to establish a continuous assessment to monitor changes (improvements or deterioration) air quality.

How can the dataset be used to disentangle sources – and here I think doing a more detailed land- use assessment for each position - all sources in vicinity of the sampling point as well as the dominant one. The overview seems to be that NO2 and NH3 are clearly the pollutants which are elevated. A transect approach across source areas may give a better view of variation which will lie on top of the kriged background maps.

I would be interested to understand the authors hypothesis for the future changes in air quality, and also how the current dataset may be exploited further (e.g. application of local/regional model to understand the pollutant source strengths required to reproduce the observations).

Conclusions section:

This section should be significantly shortened and should just give a concise overview of the main outcomes of the study. The implications of the study need to be drawn out.

---

## Author Comment (AC1) · 22 Dec 2017

**Old title: Spatial distribution of gaseous pollutants (NO2, SO2, NH3, HNO3 and O3) in Abidjan, Cote d'Ivoire.**

**New title:**

A pilot study of gaseous pollutants measurement (NO2, SO2, NH3, HNO3 and O3) in Abidjan, Cote d'Ivoire: contribution to an overview of gaseous pollution in African Cities.

December 20, 2017

Julien BAHINO Laboratoire de Physique de l'Atmosphère, Université Félix Houphouët-Boigny, Abidjan, BPV 34, Cote d'Ivoire julienbahino@gmail.com

Corinne GALY-LACAUX Laboratoire d'Aérologie, UMR 5560, Université Toulouse III Paul Sabatier and CNRS, 31400, Toulouse, France Corine.galy-lacaux@aero.obs-mip.fr

**Response to Reviewer's comments**

Dear associate Editor,

First of all, we would like to thank the reviewers for their comments. They emphasizes the importance of data reported in this paper in poorly studied areas like West Africa.

We have revised our paper taking into account all the reviewer's comments. Major modifications done in the paper in respect with the different remarks are reported below.

Please find attached a point-by-point response to reviewer's concerns.

**RESPONSE TO REVIEWERS**

**REFEREE #1**

**General comments:** This paper presents a relatively high resolution spatial distribution for inorganic gaseous species measured at Abidjan in Cote d'Ivoire. Since very little data exist for this part of Africa, I welcome any data from this region. Scientists conduct measurements under difficult conditions and results presented should be viewed within the logistical constrains associated with measurements conducted in this part of the world. However, my main concern with this paper is that results for a very short period are presented, i.e. 8 weeks during which passive sampling measurements were conducted for two-week periods. That is a very small data set, although it is considered an intensive campaign within the scope of the DACCIWA FP7 project. It would have been good if there were 8 weeks of continuous measurements, but then again the scientists would not have been able to cover the spatial distribution (21 sites). It is mentioned that one of the 21 sites is a "supersite". However, I presume that this so-called "supersite" does not contain any continuous active measurements that could have been used to present e.g. diurnal patterns.

**Reply**: We acknowledge referee #1 to recognise that our dataset representative of a relative high spatial resolution is original. Moreover, referee #1 is aware about logistical difficulties to perform this type of studies. We agree with the comments dealing with the duration of the measurement period limited to 8 weeks. It should be noted that this period, representative of the heart of the dry season has been selected as an intensive periods for the DACCIWA WP2 project. We assume that this period, even short, represents the maximum of air pollution in the city of Abidjan. Monthly variations of gaseous pollutants concentration at the 3 supersites show that maximum of concentration is always obtained between December and February. This result led the choice of study period and will be publish in the next paper.

1) I would like to suggest that the title, introduction and structure of this paper is changed. It should indicate that the main focus of this paper is to present and contextualise concentrations of inorganic gaseous species for African cities. Begin the paper with the spatial distributions measured for Abidjan, i.e. inorganic gases concentrations associated with specific source sectors. Thereafter the average concentrations of inorganic gases (maybe also include more than just NO2 and SO2) measured in Abidjan is compared to other African cities with the discussion and contextualization presented. Then you end of with comparing specific source sectors in Abidjan with other similar source sectors in other African countries. To summarize, the introduction should be re-written in order to indicate that his paper deals with concentrations of inorganic gases in African cities. You have a high spatial resolution for Abidjan where you have measured and then contextualise with other African countries.

As mentioned, my main concern with this paper is the short measurement period, i.e. 8 weeks (4 samples for each gas). I do not think that it is sufficient for publication in ACP if the paper is only presented primarily from the view of a very short sampling campaign. If the paper is pitched from a perspective to review/contextualise African cities then the impact of the paper can improve.

Reply: According to the general and specific comments included in the text, we decided:

-To change the title: A pilot study of gaseous pollutants measurement (NO2, SO2, NH3, HNO3 and O3) in Abidjan, Cote d'Ivoire: contribution to an overview of gaseous pollution in African Cities. -To restructure the introduction: we now used six paragraphs and add details on Abidjan megacity -To restructure the plan of the manuscript: see details for restructuration in section 3.2

2) The paper can be significantly shortened. In its current format, the paper is of a repetitive nature with many sentences/arguments/results repeated throughout the document. In addition, many

obvious general statements are given throughout manuscript. Restructuring the paper will also eliminate the repetitive nature of the paper.

**Reply**: The paper has been significantly shortened. The repetitive sentences, arguments and results were removed. We also removed the obvious general statements.

3) There is a lack of depth in the explanation of the observed spatial distributions. Incorporating meteorology or air mass movements, for instance, would improve the understanding and interpretations.

**Reply**: Measurements were performed at the source of air pollution to avoid the influence of the meteorological parameters such as air mass movements. This is in agreement with the strategy of the Work package 2 of the DACCIWA project where 3 sites have been instrumented (from December 2014 to March 2017). This study allows us to follow the temporal variability of inorganic gaseous pollutants. Results proposed in the supplementary Fig. S1 showed a very low seasonal variability depending on the studied gas. The monthly variation of inorganic gaseous pollutant) showed that the highest concentrations are measured during the dry season (December-February). This information led to the choice of the study period. These mid-term results will be published in another paper.

We also improved the interpretations using wind direction and aging of pollutants in air masses. (e.g.,  $HNO_3$  and  $O_3$ )

4) The paper also require significant language and text editing. Detailed comments: I have included detailed specific comments relating to the above general comments in the attached PDF version of the submitted manuscript. Many language and text edits/suggestions are indicated throughout the reviewed PDF version of the manuscript.

**Reply**: We improved the text editing trying to clearly define the paragraphs. In addition, all suggestions for rephrasing sentences and improving the level of language were taken into account.

(All comments written directly by the reviewer in the text have been taken into consideration).

**Specifics comments:**

Page 1, Line 11: Keep with the previous text *Reply: Done, the parenthesis was kept with the text of line 1.*

**Page 1, Line 13:** Inorganic gaseous, i.e. NH3, NO2, O3, SO2 and HNO3 *Reply: We replaced "Gases" by* "Inorganic gaseous, i.e. NH3, NO2, O3, SO2 and HNO3"

**Page 2, line 1:** In general I felt that this section contains one very long paragraph. I tried to indicate possible paragraphs

**Reply**: Very long paragraphs in this section were shortened according to your suggestions. Introduction was rewritten using 6 paragraphs. Parts of section 2 on the presentation of the city of Abidjan have been moved in introduction.

**Page 2, line 18**: Consider to delete this statement. The atmospheric research has increased during the last 10 years significantly

**Reply**: We agree that atmospheric research has increased during those last year in South Africa. But it should be noted that at this time, few African cities are able to provide real long term air quality monitoring. We deleted this statement

**Page 2: line 19:** What was it in 2016? It is now 2017. What was the prospect for 2017? *Reply: The GDP growth rate in 2016 is 1.4 % with a prospect of 2.9 % in 2017 (IMF, 2016)*

We changed the sentence with new data "West African region has experienced an economic upturn during the last few years characterized by an economic growth estimated at 2.7 % in 2017 with a prospect of 3.5 % in 2018 (IMF, 2017).". (now line 19-20, page 2)

Page 2, line 32: replacement Josipovic by Conradie

**Reply**: The reference Josipovic et al., 2010 has been changed by Conradie et al., 2016. (now line 31, page 1)

**Page 2, line 35:** Not be confused with the country currently called Namibia – therefore rather delete this and call it Southern West African region.

**Reply**: Done. In the DACCIWA projet "SWA" is used to indicate the south of West Africa (Knippertz et al., 2015). However, to avoid any confusion with Namibia, we called this region "Southern West African Region" as you proposed it. (now line 34, page 2)

Page 3, line 19: Ranked first in terms of what?

**Reply**: The port of Abidjan is at the first rank in West Africa in terms of merchandise traffic with 21.5 million tons in 2013 and a prospect of 27 million tons in 2018. (now line 8-9, page 3)

**Page 3, line 20- 21:** It is not very clear what is meant here. Please improve *Reply:* We decided to remove this sentence because it is already explained that Abidjan is the economic heart of West Africa.

**Page 3, line 22- 23:** What growth? Do you mean the economic growth in Cote d'Ivoire in general of West Africa? Please improve this sentence.

**Reply:** It is the annual growth rate of Cote d'Ivoire.

We changed the sentence: "The annual growth rate of GDP in Cote d'Ivoire is expected to be 9 % at the end of the third quarter of 2017. The main sectors that support the economic growth of Cote d'Ivoire are real estate, road transport, road building, manufacturing and mining industries". (now line 12-13, page 3)

**Page 3 line 24:** what does it meant with industrial zones *Reply: New zones with implantation of new industrial companies.*

**2.1 Sampling sites**

**Page 4, line 3:** These site description can be improved. I tried below with a few suggestions. Please consider revising these site descriptions to give better description of which site represent what source *Reply:* We rewrote the site description as you proposed "Figure 1 presents the mapping of the 21 selected sites referenced from  $A_1$  to  $A_{21}$  and their identified major sources of pollution nearby in the district of Abidjan while Table 1 presents details of each site ". (now line 32-34, page 3)

We also completed table 1 with description of main activities around the sampling sites. In addition we add figure 2 to present photos of some of the main instrumented sites.

Page 4, line 8: Popular district (I am not sure what is meant here)

**Reply**: It is a residential district. Population lives there with very low- income. Firewood and charcoal is used as main source of energy for cooking.

We replaced popular district by residential district. The details given in table 1 explain what the main activities around this site are.

**2.2.1 Sampling procedure:**

**Page 4, line 3:** Maybe just mention that your passive samplers were compared to active samplers and that the comparison were good or that conversion factors was necessary for certain species. **Page 4, line 21- 23:** Please improve this sentence

**Reply**: we rewrote the sentence. "Validation and inter-comparison studies have been performed according comparison with active analysers. Results presented in Adon et al 2010 assure the quality and the accuracy of the measurements". (now line 6-8, page 4)

**Page 4, line 37:** why? (2.5 meters above the ground) Because it is considered a representative height of human exposure.

**Reply**: We completed the sentence as follow "Passive samplers were mounted on stainless steel rails in duplicate and placed at a height of at least 2.5 meters above the ground corresponding to height of human exposure" (now line 27-29, page 4)

Page 4, line 39- 40: how was this calculated? Move to 2.2.2 chemical analysis section

**Reply**: To give an indication of the precision of this sampling technique, the covariance of all duplicate samples was calculated for the ten year period (1998–2007). All detail are given in the paper Adon et al., (2010). This sentence was moved from section (2.2.1) to the section (2.2.2) (now line 5-6, page 5)

**2.2.2 Chemical analysis**

A global reorganization between section 2.2.1 and 2.2.2 have been performed taken into account all the following comments

**Page 5, line 7- 9:** This sentence must be part of the previous section (2.2.1) *Reply: This sentence (coating of samplers) was moved in the section* 2.2.1

**Page 5, line 17-18:** This can be the first sentence of this paragraph or the last sentence of the previous paragraph

*Reply*: *The sentence (analytical precisions) was moved in the section (2.2.1) (now line 31-33, page 4)*

**Page 5, line 18-20:** In the previous paragraph you present detection limits calculated from field blanks. Please combine these sentence and include in this section.

**Reply**: Detection limits for samplers, analytical precision, details on calculations have been combined and now are presented in section 2.2.2. now (line 3-9, page 5)

Page 5, line 24: This is a result. Also, further on you mention 672 samplers were collected and analysed.

**Reply**: I mentioned that 672 samples were exposed and 2 were damaged. Only 670 samples were analysed and 4 of them were removed from the whole database. The final database includes 666 samples, or 94.6 % of the analysed samples. (now line 11-13, page 5)

**2.4 Spatial analysis**

We significantly changed this section with the creation of 2 subtitles to express the validity of data analysis, and cross-validation.

This reorganization is based on our new results after the analysis of the semivariograms and errors calculations (see below).

**Page 6, line 11:** Is this then a better method than kriging? Why did you use kriging in your study is this is a better method?

**Reply**: The semivariograms necessary for kriging were performed for each gas (Fig.5). Results showed that the Gaussian model of kriging was better suited for  $O_3$ ,  $HNO_3$ ,  $NO_2$  and  $SO_2$ .  $NH_3$  concentrations follow a nugget model indicating that there is no spatial autocorrelation of the data. Parameters of empirical semivariogram are available in new Table 2.

Data set analysis and cross-validation allow us to choose IDW as the interpolation method. In fact, the lowest value of the mean absolute error (MAE) and mean squared error (RMSE) and the highest coefficient regression value ( $R^2$ ) were found for IDW (Table 3).

**Page 6, line 17-18:** If there was a good correlation between calculated and interpolated values, I would expect that the slope of the linear equation must be closer to one. If you look at the NO2, for instance, an interpolated value of ~7 ppb correspond to an observed value of ~3 ppb. Therefore R2 is in this case not a good indicator of the correlation between interpolated and observed. The O3 looks good.

**Reply:** Comparison between IDW and kriging indicates that best correlation coefficient ( $R^2 = 0.999$ ) were obtained with IDW for all the pollutants (Fig.6). Other statistical parameters such as mean absolute error (MAE) and root mean square error (RMSE) were also determined. Results are now presented in Table 3

Page 6, line 28-30: Did you test the IDW method?

**Reply**: Yes we did. The cross-validation showed that IDW is the best interpolator for our data.

**3** Results and discussion**

Page 6, line 32: Repeating. Just write during the "study period".Page 6, line 32-33: This was also given in the Experimental section. I suggested that it is should either be reported here (Results) or in the Experimental Section

Page 6, line 34-35: This is basic information that must be in the experimental section.

Reply: we moved these sentences (line 32 to 35) in the experimental section:

**Page 6, line 35:** These two sentences can just be replaced with "In Table 2 the mean concentrations measured at the 21 sites are presented." I also think it is not necessary to include the mean concentrations in a table format. I suggest that the mean concentrations are included in the each of the figures, which can make these values available for modellers. These differences between the sites are discussed in the spatial distribution discussion that follows. The first four sentences in the Results section can be excluded.

**Reply**: We replaced the two sentences by that you proposed "In Table 4 the mean concentrations measured at the 21 sites are presented."

We decided to include the concentration measured each two-week period in the supplementary table S1 as proposed by referee #2. The average concentration measured at each site during the study period is also now included in each figure (maps).

Table 2 becomes Table 4.

We added in table 4 the minimum, maximum and average concentrations of each pollutant at each site as proposed by referee #2.

We removed the first four sentences from this part.

**Page 6, line 36-40:** These sentences are also not necessary for the paper. *Reply: We removed these sentences from the paper*

**Page 7, line 7:** For me the short on the long of Fig. 6 here is that NO2 concentrations are highest in the city center where there are more vehicles.

**Reply:** We agree with this comment and we argue in our interpretation that  $NO_2$  concentrations follow the road map in the city of Abidjan (see figure 7 where green lines represent the main road).

**Page 7, line 16-17:** What similar results were found for Los Angeles. Also, why compare to a North American site in 1981? Is'nt there something more recent to cite? *Reply: We removed this sentence because it is a non-appropriate comparison.*

**Page 7, line 26-28**: Here you relate BC to NO2 emissions. Please explore on the relevance of this. *Reply*: we decided to remove this sentence. Initially the idea was to do the link with a future ACP paper (same DACCIWA special issue) on emissions factors but the link is not direct using a particulate BC emission factor compared to surface gas concentration.

**Sulfur dioxide**

**Page 8, line 1:** According to the figures shown here, traffic to me seems like a less important sources of SO2 for Abidjan, which is usually the case for SO2. Also, SO2 concentrations seem low near A11 where lack of desulfurization is given a possible source. What type of industries are located here? There must be another explanation for the SO2 distribution observed here, i.e the peak in the north-eastern sector. Have you considered meteorological reasons e.g. air mass movements?

**Reply**: We think that considerations about combustion sources explain partly the concentrations levels presented on the map for traffic and industrial sites. But as the reviewer mentions, it seems also that the north east gradient could be also related to air masses circulation in the dry season with north east harmattan winds.

**Page 8, line 15-20:** In South Africa SO2 is not usually associated with domestic burning. Why do you think this is the case for Abidjan? *Please see answer below*

**Ammonia**

**Page 8, line 22:** Is this true? I am not so sure. Maybe say that it is specific for tropical regions. This is most certainly not the case for southern Africa, for example.

**Reply**: yes, we think that it is a typical feature for tropical African regions where biomass burning and biofuel combustion are predominant and influence air quality at the urban scale. Below we give you some references on biomass burning and biofuel combustion as the main source of NH3. (added in the paper).

Akagi, S. K., Yokelson, R. J., Wiedinmyer, C., Alvarado, M. J., Reid, J. S., Karl, T., Crounse, J. D. and Wennberg, P. O.: Emission factors for open and domestic biomass burning for use in atmospheric models, Atmospheric Chem. Phys., 11(9), 4039–4072, doi:10.5194/acp-11-4039-2011, 2011.

Paulot, F., Paynter, D., Ginoux, P., Naik, V., Whitburn, S., Van Damme, M., Clarisse, L., Coheur, P.-F. and Horowitz, L. W.: Gas-aerosol partitioning of ammonia in biomass burning plumes: Implications for the interpretation of spaceborne observations of ammonia and the radiative forcing of ammonium nitrate, Geophys. Res. Lett., 44(15), 2017GL074215, doi:10.1002/2017GL074215, 2017.

Whitburn, S., Van Damme, M., Kaiser, J. W., van der Werf, G. R., Turquety, S., Hurtmans, D., Clarisse, L., Clerbaux, C. and Coheur, P.-F.: Ammonia emissions in tropical biomass burning regions: Comparison between satellite-derived emissions and bottom-up fire inventories, Atmos. Environ., 121(Supplement C), 42–54, doi:10.1016/j.atmosenv.2015.03.015, 2015.

**Page 8, line 24:** I am not so sure of NH3 emissions from traffic. Could you please give specific references for this? NH3 is usually associated with agriculture and bacterial decomposition of animal excreta. Maybe seabird colonies can be source on Abidjan.

**Reply**: We agree with referee #2 that  $NH_3$  is usually associated with agriculture and bacterial decomposition of animal excreta. Some studies have also reported NH3 emission from traffic. Please find below some specific references (added in the paper).

(Sun et al., 2017; Prevot et al., 2017; Teng et al., 2017; Liu et al., 2014; Perrino et al., 2002; Kean et al., 2000)

Kean, A. J., Harley, R. A., Littlejohn, D. and Kendall, G. R.: On-Road Measurement of Ammonia and Other Motor Vehicle Exhaust Emissions, Environ. Sci. Technol., 34(17), 3535–3539, doi:10.1021/es991451q, 2000.

Liu, T., Wang, X., Wang, B., Ding, X., Deng, W., Lü, S. and Zhang, Y.: Emission factor of ammonia (NH 3) from on-road vehicles in China: tunnel tests in urban Guangzhou, Environ. Res. Lett., 9(6), 064027, doi:10.1088/1748-9326/9/6/064027, 2014.

Perrino, C., Catrambone, M., Di Menno Di Bucchianico, A. and Allegrini, I.: Gaseous ammonia in the urban area of Rome, Italy and its relationship with traffic emissions, Atmos. Environ., 36(34), 5385–5394, doi:10.1016/S1352-2310(02)00469-7, 2002.

Prevot, A. S. H., Elser, M., El Haddad, I., Maasikmets, M., Bozzetti, C., Robert, W., Richter, R., Slowik, J., Teinemaa, E., Hueglin, C. and Baltensperger, U.: Spatial variability and sources of ammonia in three European cities, vol. 19, p. 11902. [online] Available from: http://adsabs.harvard.edu/abs/2017EGUGA..1911902P (Accessed 8 October 2017), 2017.

Sun, K., Tao, L., Miller, D. J., Pan, D., Golston, L. M., Zondlo, M. A., Griffin, R. J., Wallace, H. W., Leong, Y. J., Yang, M. M., Zhang, Y., Mauzerall, D. L. and Zhu, T.: Vehicle Emissions as an Important Urban Ammonia Source in the United States and China, Environ. Sci. Technol., 51(4), 2472–2481, doi:10.1021/acs.est.6b02805, 2017.

Teng, X., Hu, Q., Zhang, L., Qi, J., Shi, J., Xie, H., Gao, H. and Yao, X.: Identification of major sources of atmospheric NH3 in an urban environment in northern China during wintertime, Environ. Sci. Technol. [online] Available from: http://pubs.acs.org/doi/abs/10.1021/acs.est.7b000000000328 (Accessed 8 October 2017), 2017.

**Page 9, line 4-6: Very general. Please improve this sentence.**

**Reply**: The sentence has been modified: "In addition to the breedings, this urban area combines large industrial zones and individual plantations of hevea and banana. These plantations often make excessive use of chemical fertilizers rich in urea and leading to ammonia emissions (Degré et al., 2001"). (now line 29-31, page 8).

**Page 9, line 9-11:** This is the main finding here. NH3 is very high, which can be attributed to domestic burning, as well as bacterial decomposition associated with agriculture (pig and poultry farming) and waste/garbage dumps.

**Reply:** We rewrote the sentence as you proposed. "To summarize,  $NH_3$  concentrations are very high, which can be attributed to domestic fires, as well as bacterial decomposition associated with agriculture (pig and poultry farming) and waste dumps. (now 35-36, page 8).

**Nitric acid**

Page 9, line 13: very general statement *Reply: We removed this sentence*

**Page 9, line 17-24:** HNO3 is very difficult to explain. Can maybe related to aged air masses, i.e. NO2 emissions from the city center being transported mainly in a easterly direction. Need to bring in meteorology.

**Reply**: South-West prevailing winds direction and aged air masses were used to explain highest concentrations at the East of the city.

**Ozone**

**Page 9, line 25:** In general, the O3 concentrations are very low. Have you compared passive O3 concentrations with active O3 concentrations? From your spatial distribution maps, O3 seems to be the lowest on an easterly direction from the central city. High NO2 is usually associated with lower O3 due to titration. Therefore, lower O3 is expected over the city. It seems to me that the highest O3 concentrations were measured at sites in the east the furthest from the city center. You must look at wind direction and/or air mass history to indicate predominant air mass movement.

**Reply**: Active ozone analyser (Thermo Scientific Model 49i Ozone Analyser) have been installed in the suburban site ( $A_{17}$ ) of Bingerville where maximum of concentration is expected.

The main objective was first to study the diurnal variation of ozone concentration from December 2014 to April 2017. Results over the study period show comparable concentrations with passive samplers. Results will be published in the next paper. South-West prevailing winds direction and aged air masses were used to explain highest concentrations at the East of the city.

Page 9, line 27-29: very general statement.

**Reply**: We rewrote and simplified the sentence " $O_3$  is often associated in urban areas to adverse effects on human health and natural environment. (Hagenbjörk et al., 2017). It is well known that this secondary pollutant depends on photochemical interactions of gaseous precursors composed of nitrogen oxides (NOx) and volatile organic compounds (VOCs) (Duan et al., 2008; WHO, 2006). (now line 8-10, page 9)

Page 10, line 5-7: Why would one associate high O3 wit waste burning sites?

**Reply**: Higher Ozone concentrations are associated to the waste burning sites because this site is located far from de city center.

**3.2** Mean gaseous concentrations of the main anthropogenic activities in Abidjan and comparison with other cities of developing countries**

We significantly change this section with the creation of 2 subtitles

3.2.1 Average concentration of primary gaseous pollutants associated with anthropogenic activities Only the primary pollutants (NO2, NH3 and SO2) were associated to specific source of pollution.

3.2.2. Average concentrations of gases compared with other urban sites We compared mean concentrations in Abidjan with other concentrations in the world measured in urban sites and traffic on the one hand and suburban sites on the other hand.

**Page 10, line 9-10:** In this section the sources characterisation must be better explained in the Experimental section. How do you differentiate between domestic fires, residential and suburban? This entire section must be rewritten in view of an improved source characterisation.

**Page 10, line 15**: I cannot see how you can differentiate between residential and domestic burning. Also, what is the difference between residential and suburban?

The source characterisation must be better explained in the experimental section and redefined in order to improve this entire section.

**Reply**: Domestic fires is a site where charcoal and firewood are mainly used as source of energy. Residential is a site where people lives with low or middle income. LPG, charcoal and firewood can be used as a source of energy for cooking.

Suburban site is located far from the city center (more than 15 km)

Table 1 now improves these different definitions of sites and gives more information on sources.

What about waste decomposition for NH3?

**Reply**: Waste decomposition and human waste management have also been identified as a potential source of  $NH_3$  (now line 29-33, page 8).

**Page 10, line 16:** I would expected traffic emissions to be significantly higher *Reply: Industrial sites are strongly influenced by traffic. They combine Traffic and other sources of pollution. The average concentration in industrial sites*  $(20.9 \pm 2.8ppb)$  *is slightly higher than Traffic sites*  $(17.8 \pm 4.7 ppb)$ .

**Page 10, line 25:** It does not make sense to relate O3 levels with specific types of sites. O3 is the highest in an easterly direction far from the city center.

**Reply**: Higher Ozone concentrations are associated to the waste burning site because this site is located at the East far from de city center. In addition, south west prevailing winds suppose higher concentration in the easterly direction.

Page 10, line 32: Waste decomposition, I suppose? *Reply:* Waste burning site combines waste decomposition and waste combustion

**Page 11, line13-30:** You are not comparing apples with apples and pears with pears in these paragraphs. Inorganic gaseous concentrations associated for instance with traffic at Abidjan must be related to concentrations of inorganic gases at traffic sites in other parts of the world.

This contextualisation is important, but this discussion must be improved.

Also, the Amersfoort site used in Table 3 does not make sense for Abidjan where inorganic gaseous concentrations were measured for different sectors within a city. Amersfoort is a regional site. Again, compare apples with apples.

*Reply:* We significantly change this section with the creation of 2 subtitles

3.2.1 Average concentration of primary gaseous pollutants associated with anthropogenic activities Only the primary pollutants ( $NO_2$ ,  $NH_3$  and  $SO_2$ ) were associated to specific source of pollution and presented in Fig. 12

3.2.2. Average concentrations of gases compared with other urban sites We compared mean concentrations in Abidjan with other concentrations in the world measured in urban sites and traffic on the one hand and suburban sites on the other hand.

**3.3 Overview of urban NO2 and SO2 monitoring studies in African cities**

Page 11, line 31: I would like to suggest that the title and introduction of this paper is changed.

It should indicate that the main focus of this paper is to present and contextualise concentrations of inorganic gaseous species for African cities. You then begin the paper with the spatial distributions measured for Abidjan, i.e. inorganic gases concentrations associated with specific source sectors. Thereafter the average concentration of inorganic gases (maybe also inlcude more than just NO2 and SO2) measured in Abidjan is compared to other African cities and the discussion presented. Then you end of with comparing specific source sectors in Abidjan with other similar source sectors in other African countries.

To summarize, the introduction should be re-written in order to indicate that his paper deals with concentrations of inorganic gases in African cities. You have a high spatial resolution for Abidjan which you measured and then contextualise with other African countries.

My main concern with this paper is the short measurement period, i.e. 8 weeks (4 samples for each gas). I do not think that it is sufficient for publication in ACP if the paper is only presented primarily from the view of a very short sampling campaign. If the paper is pitched from a perspective to review/contextualise African cities then the ims+pact of the paper can improve.

Restructurings will also eliminate the repetitive nature of the paper.

**Reply**: According to the general and specific comments included in the text, we decided as previously said:

To change the title: A pilot study of gaseous pollution measurement ( $NO_2$ ,  $SO_2$ ,  $NH_3$ ,  $HNO_3$  and  $O_3$ ) in Abidjan, Cote d'Ivoire: comparison with other African cities. To restructure the introduction: we now used six paragraphs and add details on Abidjan megacity. To restructure the plan of the manuscript:

Page 12, line 2-3: This information contained in the figure, which is not necessary to repeat in the text

**Page 12, line 11-13:** This must be part of caption and not included in text. *Reply:* As you proposed, this information already present in figure caption was removed from the text.

**Page 12, line 17:** Theses two sections (SO2 and NO2) can be improved. *Reply: We improved the text editing trying to clearly define the paragraphs*

**4 Conclusion and recommendations**

Page 13, line 25: To be re-written within view of new scope of paper.

**Reply**: This section was rewritten within the new scope of the paper and also significantly shortened as proposed by the both referees

The conclusion has been shortened highlighting the main results of the pilot experiment to measure gaseous pollution in Abidjan. In addition, one paragraph is added (line 17 to 27) on the future projects in Cote d'Ivoire planned for 2018 both from a research and operational point of view. In this context, we try to explain that all the results presented in this work will help in the development of these new projects and will serve to develop the experimental strategy of such a network. In addition, meteorological parameters will be included to finally lead to modeling studies to predict air quality for Africa to regional to urban scale.

**REFEREE #2**

**Overview:**

It is really welcome to see a paper with new chemically detailed air quality pollutant data from Abidjan, Cote d'Ivoire. As the authors outline there is currently very little information in this region and hence the paper represents a step change in knowledge. The protocols for the passive samplers are well applied with excellent quality assurance protocols. However in order to be sufficient to publish in ACPD, there needs to be a clearer detailing of all the data and a stronger interpretative section in the paper. There also needs to be a detailed review of the language and word use throughout as some of the choices of words do not fit and many sentences need to be sense checked.

**Reply**: We acknowledge referee #2 to recognise that protocols using passive samplers are well applied with excellent quality assurance protocols.

All the data for each measurement period have been clearly detailed in a supplementary Table S1. Data presented in the overview have also been added in 2 supplementary tables (Table S2 for NO2 and Table S3 for SO2).

We have tried also to improve the explanations and language. We improved the text editing trying to clearly define the paragraphs

**Technical and scientific comments:**

**Introduction**

**Line 32 onwards:** There is a lot of discussion of the specific project through which the research was done which is not particularly relevant for the paper. Could this be revised to be shorter and more specific to the intensive long term sites – what is being measured at them and at what resolution – if the supersite in Abidjan has been operational for the year it would be useful to know more detail about that and how the intensive reported fits in. Beyond a primary survey of background air quality pollutant concentrations, what are the starting hypotheses and how does this study address them?

**Reply**: The introduction has been restructured according 6 paragraphs. (see comments referee #1) Mid-term measurements sites are operational. Measurements were carried out bimonthly from December 2014 to March 2017. The seasonal variation of gaseous pollutants showed that the highest concentrations are measured during the dry season (December-February). This information led to the choice of the study period.

**Experimental design**

**P3 Line 9:** onwards the economic description of Abidjan should really be in the introduction as it does not relate to the experiment. This paragraph could be shortened and refer to Figure 1 perhaps combining with section 2.1

**Reply**: The economic description of Abidjan has been shortened and reported in the introduction as recommended by the both referees.

**P3 line 36:** Is the Tecsult report publically available or could it be put in the supporting material? If not then rather than refer to it, the details of the report suggested network design should be in the details of this paper, and perhaps can be critiqued. . .did the network design meet the aims of the project? Can you extrapolate from this number of sites to the whole region for all pollutants (my feeling is that for ozone it may be fine, but for ammonia it may not be the best approach). Referencing the global literature on kriging and air quality impact assessment would be good. A variogram fit would be useful to understand how well the kriging worked for the different pollutants. What is the topography of the area?

**Reply**: Tecsult report is not publically available. All references to this report were removed from the paper.

For the interpolation between the sites to build spatial maps: we add in the paper the figure 5 showing the empirical semivariogram for each pollutant. The results showed that the Gaussian kriging model was suitable for  $O_3$ ,  $HNO_3$ ,  $SO_2$  and  $NO_2$ .  $NH_3$  semivariogram showed a nugget model .As the reviewer thoughts, we found that there is no spatial autocorrelation with  $NH_3$  data. We can not interpolate  $NH_3$  data using kriging.

Data set analysis and cross-validation allow us to choose IDW as the interpolation method (figure 6). In fact, the lowest value of the mean absolute error (MAE) and mean squared error (RMSE) and the highest coefficient regression value ( $R^2$ ) were found for IDW.

We also added the measured values on all the maps as proposed by referee #1.

The altitude of the city varies gradually between o m at the Atlantic coast (south) and 110 m north of the city. It does not contain mountains. It is surrounded by vegetation composed of agricultural crops. Table below gives elevation at each location site.

| Sites | Elevation (m) | Sites | Elevation (m) | Sites | Elevation (m) |
|-------|---------------|-------|---------------|-------|---------------|
| A1    | 31            | A8    | 110           | A15   | 46            |
| A2    | 70            | A9    | 83            | A16   | 39            |
| A3    | 11            | A10   | 19            | A17   | 20            |
| A4    | 56            | A11   | 10            | A18   | 31            |
| A5    | 33            | A12   | 50            | A19   | 23            |
| A6    | 2             | A13   | 101           | A20   | 6             |
| A7    | 46            | A14   | 4             | A21   | 1             |

P4 line 4: What method was used to assign the land use type for each station?

**Reply**: In most of the cases, the main sources of pollution around the sites have been used to assign land uses. The information in the table below gives the definition of the type of land use.

| Land use type  | Method to assign land use                                                                                                                                                              |
|----------------|----------------------------------------------------------------------------------------------------------------------------------------------------------------------------------------|
| Domestic fires | Site where charcoal and firewood are mainly used as source of energy                                                                                                                   |
| Residential    | Site in which housing predominates. This includes private residences and community housing. People live there with low or mid-income. LPG, charcoal and firewood are used for cooking. |
| Suburban       | Residential district located on the outskirts of the city (around 15 km from the city center). Firewood and Charcoal are used for cooking                                              |
| Traffic        | The traffic sites are not far from a main road (highway). These sites are also characterized by traffic jams and a mixture of vehicles                                                 |
| Industrial     | Sites located in an industrial area.                                                                                                                                                   |
| Waste burning  | Waste burning site combines waste decomposition and waste combustion.                                                                                                                  |

**P4 line 39:** It would be useful to discuss potential interferents/artefacts: e.g. HNO3 will have interferents from other reactive oxidised N species, and NH3 potentially amines – depending on the coating type. Could the authors add a little detail on this, even though they are in the reference Adon (2010) I think it is worth re-iterating them in this paper, perhaps in a table.

**Reply**: As the referee # 2 said, we are aware that using passive samplers, it exists a potential interference between Nitric acid and  $NO_2$ . Potential interferences have been studied in the paper Adon et al., (2010).

*In the analytical process of the nitric acid passive sampler, we can have an idea of this interference. First, concerning the trapping of these two gases, i.e,* NO2 *and* HNO3*:*

The presence of sodium hydroxide (NaOH) in the impregnation solution aims to maintain a strongly basic pH (pH> 12) and thus limits the oxidation of nitrite ion  $NO_2^-$  to nitrate ion  $NO_3$  In addition, we know that NaOH molecules react with atmospheric  $CO_2$  to form water molecules that favour (or further)  $NO_2$  retention on the filter. Then, the choice of this basic solution allows simultaneous to capture on Whatman filter other acid gases such as HNO3.

Secondly, concerning the analysis of these two samplers in ionic chromatography:

Nitrate ions  $NO_3^-$  were detected in IC for the  $NO_2$  filters analysis with very low values of ppb. We only use nitrite  $NO_2^-$  ions results to estimate the concentrations of gaseous  $NO_2$ .

In the case of  $HNO_3$  filters, nitrite ions  $(NO_2^{-1})$  were not detected or at least, they are below the detection limit. So, in our opinion, the concentrations of  $HNO_3$  do not suffer of too much interference from  $NO_2$  or we assume that this interference is negligible.

To give an order of magnitude of the ratio between nitrite  $(NO_2^{-1})$  and nitrate  $(NO_3^{-1})$  ions detected in IC for NO2 filters, supplementary Table S1 indicates results for the traffic site  $(A_1)$  IC results in  $\mu g/L$ , concentration in air (ppbv) and the ratio in %. This ratio is estimated around 2% (ppbv air).

Table S 1: Ratio between nitrite  $(NO_2^-)$  and nitrate  $(NO_3^-)$  ions detected in IC for NO2 filters. Results for Abidjan Traffic site  $(A_1)$

|             | Concentration (µg/L) |                      |           | Concentration (ppbv) |                      |           |
|-------------|----------------------|----------------------|-----------|----------------------|----------------------|-----------|
| NO2 sampler | NO2_NO2 - | NO2_NO3 - | Ratio (%) | NO2_NO2 - | NO2_NO3 - | Ratio (%) |
| SA1 1016 f  | 752                  | 18                   | 2.39      | 10.23                | 0.19                 | 1.88      |
| SA1 1016 f  | 747                  | 10                   | 1.34      | 10.16                | 0.08                 | 0.81      |
| SA1 1116 d  | 937                  | 22                   | 2.35      | 11.16                | 0.22                 | 1.94      |
| SA1 1116 d  | 964                  | 21                   | 2.18      | 11.48                | 0.20                 | 1.78      |
| SA1 1116 f  | 932                  | 13                   | 1.39      | 11.95                | 0.12                 | 0.97      |
| SA1 1116 f  | 906                  | 44                   | 4.86      | 11.62                | 0.52                 | 4.45      |
| SA1 1216 d  | 1058                 | 36                   | 3.40      | 13.45                | 0.44                 | 3.24      |
| SA1 1216 d  | 1214                 | 46                   | 3.79      | 15.45                | 0.56                 | 3.65      |
| SA1 1216 f  | 1327                 | 18                   | 1.36      | 16.99                | 0.21                 | 1.21      |
| SA1 1216 f  | 1401                 | 27                   | 1.93      | 17.94                | 0.32                 | 1.79      |
| SA1 0117 d  | 1199                 | 23                   | 1.92      | 14.24                | 0.25                 | 1.76      |
| SA1 0117 d  | 1179                 | 23                   | 1.95      | 14.00                | 0.25                 | 1.79      |
| SA1 0117 f  | 999                  | 17                   | 1.70      | 12.15                | 0.18                 | 1.51      |
| SA1 0117 f  | 1002                 | 20                   | 2.00      | 12.19                | 0.22                 | 1.81      |
| SA1 0217 d  | 1236                 | 22                   | 1.78      | 15.50                | 0.13                 | 0.82      |
| SA1 0217 d  | 1274                 | 23                   | 1.81      | 15.99                | 0.14                 | 0.88      |
| SA1 0217 f  | 774                  | 29                   | 3.75      | 11.28                | 0.26                 | 2.26      |
| SA1 0217 f  | 623                  | 27                   | 4.33      | 9.02                 | 0.23                 | 2.50      |
| SA1 0317 d  | 723                  | 25                   | 3.46      | 9.05                 | 0.17                 | 1.85      |
| SA1 0317 f  | 820                  | 29                   | 3.54      | 8.13                 | 0.17                 | 2.13      |
| SA1 0417 d  | 755                  | 22                   | 2.91      | 10.91                | 0.15                 | 1.36      |
| SA1 0417 d  | 672                  | 23                   | 3.42      | 9.68                 | 0.16                 | 1.69      |
| Mean        | 727.8                | 25.8                 | 3.6       | 9.7                  | 0.2                  | 2.0       |

We decided to include 2 sentences on potential interferences in the text. We also added results for the traffic site  $A_1$  in Table S1 as a supplementary document.

"Potential interferences between HNO3 and NO2 passive samplers were estimated and found to be negligible. For the traffic site  $A_1$ , our results presented in supplementary Table S1 show that nitrate  $(NO_3^-)$  concentration are very low compared to nitrite  $(NO_2^-)$  ions with a ratio around 2 %." (line 9-12, page 5)

**Section 2.2.2** I am impressed by the protocols used for the passive samplers. The blanks are slightly higher than I would expect for ammonia field blanks but data capture was good. The purpose of the kriging needs to be put in context a bit more clearly: Is the purpose to derive background maps onto which high resolution data (e.g. from Supersite) can be superimposed? Despite the high correlations, the use of emission dominated sites (e.g. the traffic sites) is likely to skew the background map high. Ideally all sites should be away from a specific strong source – however this may be what has actually done – Table 1 needs to explain the land use categories. It is noted that the advantage of measurement based concentration maps is that the avoid emission factor errors, but the drawbacks also need to be referenced.

**Reply**: We also completed Table 1 with description of main activities around the sampling sites.

**P6: Section 3.1** It is disappointing that having had several exposure periods the authors went directly to using the average and also comparing the average of the period (2 months) to an annual average. Given that many of the pollutants have seasonal cycles as well as shorter term cycles some discussion about the appropriateness of extrapolating to an annual average is necessary. Also it would be useful to see the data for each site for each measurement period. Perhaps a whisker plot with the average max and min or just a time series for each site and chemical?

**Reply**: Measurements were performed at the source of air pollution to avoid the influence of the meteorological parameters such as air mass movements. This is in agreement with the strategy of the Work package 2 of the DACCIWA project where 3 sites have been instrumented (from December 2014 to March 2017. This study allows us to follow the temporal variability of inorganic gaseous pollutants. Results proposed in the supplementary Fig. S1 showed a very low seasonal variability depending on the studies gas. Mid-term measurements were also carried out bimonthly from December 2014 to March 2017. The monthly variation of inorganic gaseous pollutants showed that the highest concentrations are measured during the dry season (December-February). This information led to the choice of the study period. This results will be published in another paper.

We added in supplementary Table S1 all the data for each site and for each measurement period. We also completed Table 2 (now Table 4) with minimum, maximum and average concentration at each site.

**NO2 section P7 Line 12:** Comparison of short term data against the annual average is not useful. Is it expected that the NO2 is highest in the dry season or are emissions approximately constant over the year? If extrapolating to an annual average, the assumptions made should be made clear.

**Reply**: We agree with the referee that  $NO_2$  concentration in the dry season can be extrapolate to the year.

In agreement with the strategy of the Work package 2 of the DACCIWA project where 3 sites have been instrumented (from December 2014 to March 2017). This study allows us to follow the temporal variability of inorganic gaseous pollutants. Results proposed in the supplementary Fig. S1 showed a very low seasonal variability depending on the studies gas. The monthly variation of inorganic gaseous pollutants showed that the highest concentrations are measured during the dry season (December-February). This information led to the choice of the study period. These results will be published in another paper.

**P7 line 24**: The part about black carbon in this section does not appear to fit with the discussion around NO2. It either needs to be removed or a discussion of black carbon correlation with NO2 from the Keita pers comm reference. If it is a pers comm, perhaps the data could be added to this paper?

**Reply**: We decided to remove this sentence. Initially the idea was to do the link with a future ACP paper (same DACCIWA special issue) on emissions factors but the link is not direct using a particulate BC emission factor compared to surface gas concentration.

**Ammonia**

**P8 line 21** Human waste (and its management) is not really mentioned as a source of ammonia though it is likely to be a significant source.

**Reply**: We agree with you that human waste and its management can be a source of NH3. We add some references in the text. (line 32-33, page 8)

**P8 line 34** The measurements 50 m from sources is appropriate. Could the authors clarify how close the other measurement points were from NH3 sources. Again the full dataset should be presented in supplementary material so that readers can understanding the details rather than just the range being presented.

**Reply**: We added as supplementary materials All the measurements of each pollutant at each site. Table S1 All the NO2 and SO2 concentration presented in the overview Table S2 and Table S3

**P9 line 9** at the end of the ammonia section the first part is repeated. Some tightening of the text would improve the manuscript here Nitric acid *Reply: The repetition at the end of this was removed*

**P9 line 20:** the statement that "HNO3 appears to be emitted from several sources" cannot be backed up by the measurements. HNO3 as the authors stated is frequently a secondary pollutant, rather than a primary emission. Also there are several other gas phase oxidised nitrogen chemicals which could also contribute to a passive HNO3 sampling so some discussion of potential other contributions and reference to a review or overview of the issue.

*Reply*: We agree with you that HNO3 is a secondary pollutant.

This statement was removed from the text.

Some discussions focus on  $HNO_3$  precursor reactions. The wind direction was also used to explain the high concentration of  $HNO_3$  in the northwest of the city.

**P9 line 22:** I think there are papers in the literature which have made direct observations of HNO3 and therefore it would be useful to discuss the results in terms of the literature (although there is some discussion later)

**Reply**: We agree with you that it would be useful to discuss the concentration of  $HNO_3$  in terms of literature. However, very few studies on the direct observation of HNO3 in urban areas in Africa have been found.

**Section 3.2** The discussion of the different source attributions would be improved by a more detailed description of each of the sites – perhaps a focused attribution for NO2 using smaller maps or photos. This section is useful but slightly hard to follow with just the maps and average concentrations. Significant amounts of the text are fairly similar to the preceding sections on the measurements.

**Reply**: We significantly change this section with the creation of two subsections 3.2.1 Average concentration of primary gaseous pollutants associated with anthropogenic activities Only the primary pollutants ( $NO_2$ ,  $NH_3$  and  $SO_2$ ) were associated to specific source of pollution. 3.2.2. Average concentrations of gases compared with other urban sites We compared mean concentrations in Abidjan with other concentrations in the world measured in urban sites and traffic on the one hand and suburban sites on the other hand.

More details on sites description was given in Table 1. Some photo of DACCIWA sampling sites are presented in Fig.2.

**Section 3.3** The overview of the urban studies across the African continent is very useful. However it would be useful for the authors to adjust the discussion from listing the measurements and the range of values, but also to consider the impacts and purpose of the measurements. The snapshot campaign provides a picture, and it seems that Abidjan sits in a similar range to many other cities. A short atmospheric chemistry section discussing the mix of the pollutants and hence if one was to establish a continuous assessment to monitor changes (improvements or deterioration) air quality.

How can the dataset be used to disentangle sources – and here I think doing a more detailed land- use assessment for each position - all sources in vicinity of the sampling point as well as the dominant one. The overview seems to be that NO2 and NH3 are clearly the pollutants which are elevated. A transect approach across source areas may give a better view of variation which will lie on top of the kriged background maps.

I would be interested to understand the authors hypothesis for the future changes in air quality, and also how the current dataset may be exploited further (e.g. application of local/regional model to understand the pollutant source strengths required to reproduce the observations).

**Reply**: We listed the measurements of  $NO_2$  and  $SO_2$  across African cities in 2 supplementary tables (Table S2 and Table S3, respectively). Minimum, maximum and average concentrations are also given.

We agree with referee #2 that Abidjan sits in a similar range to many other cities, as discussed in the Section 3.3 and the conclusion.

Hypothesis for the future changes in air quality and the possible exploitation of the dataset are included in the conclusion for the future projects.

**Conclusions section:** This section should be significantly shortened and should just give a concise overview of the main outcomes of the study. The implications of the study need to be drawn out.

**Reply**: The conclusion has been shortened highlighting the main results of the pilot experiment to measure gaseous pollution in Abidjan. In addition, one paragraph is added (line 17 to 27) on the future projects in Cote d'Ivoire planned for 2018 both from a research and operational point of view. In this context, we try to explain that all the results presented in this work will help in the development of these new projects and will serve to develop the experimental strategy of such a network. In addition, meteorological parameters will be included to finally lead to modeling studies to predict air quality for Africa to regional to urban scale.

We sincerely hope that you will consider our responses and modifications of the paper as acceptable. With many thanks,

Regards,

Julien Bahino and Corinne Galy-Lacaux

---

## Author Comment (AC2) · 22 Dec 2017

Please find attached supplementary materials

Please also note the supplement to this comment:
https://www.atmos-chem-phys-discuss.net/acp-2017-724/acp-2017-724-AC2-supplement.pdf
* * *